**Investigation**

# Roles and regulation of the Kunitz domain protein MLT-11 during *C. elegans* cuticle synthesis and molting

James Matthew Ragle (iD),[1] Ariela Turzo,[1] Anton Jackson,[1] An A. Vo (iD),[1] Vivian T. Pham,[1] Keya Daly (iD),[1]
John C. Clancy (iD),[1] Max T. Levenson,[1] Alex D. Lee (iD),[1] Jordan D. Ward (iD)[1,*]

[1]Department of Molecular, Cell, and Developmental Biology, University of California-Santa Cruz, Santa Cruz, CA 95064, United States

*Corresponding author: Department of Molecular, Cell, and Developmental Biology, University of California-Santa Cruz, Santa Cruz, CA 95064, United States. Email: jward2@ucsc.edu

Apical extracellular matrices (aECMs) are associated with many epithelia and often form a protective layer against biotic and abiotic threats in the environment. Despite their importance, we lack a deep understanding of their structure and dynamics in development and disease. *Caenorhabditis elegans* molting offers a powerful entry point to understanding developmentally programmed aECM remodeling. Here, we show that the poorly characterized putative protease inhibitor gene *mlt-11* is directly regulated by the NHR-23 transcription factor. We identify key *cis*-regulatory elements required for robust *mlt-11* expression. An internal MLT-11::mNeonGreen translational fusion transiently localized to the aECM in the cuticle and embryo. MLT-11::mNeonGreen also lined openings to the exterior (vulva, rectum, and mouth). *mlt-11* is necessary to pattern all layers of the adult cuticle, and reduction of MLT-11 levels disrupted the barrier function of the cuticle. Deletion of conserved Kunitz protease inhibitor domains or intervening sequences produced a range of defects including either left or right roller phenotypes and small separations of the cuticle along the length of the animal (microblisters). MLT-11 is processed into at least 2 fragments, and internal and C-terminal mNeonGreen knock-ins display distinct localization patterns. Predicted *mlt-11* null mutations caused fully penetrant embryonic lethality and elongation defects. Together, this work suggests that MLT-11 localizes similarly to precuticle components, and conserved sequences play distinct roles in promoting proper assembly of the aECM.

Keywords: *C. elegans*; molting; gene regulation; protease inhibitor; apical extracellular matrix; nuclear hormone receptor; embryogenesis; Animalia

## Introduction

Specialized extracellular matrices cover the apical surface of most epithelial cells and form the skin in almost all animals (Bonnans et al. 2014; Zheng et al. 2020; Diller and Tabor 2022). These apical extracellular matrices (aECMs) also line the lumen of internal tubular epithelia to form a protective layer against biotic and abiotic threats (Bonnans et al. 2014; Zheng et al. 2020; Diller and Tabor 2022). Despite their importance, understanding the structure and dynamics of aECM components in development and disease remains challenging.

*Caenorhabditis elegans* is a powerful model to study aECM structure and remodeling. Animals have a collagen-based aECM (cuticle) that may provide insight into general skin biology and its dynamics (Page and Johnstone 2007; Sundaram and Pujol 2024). The components of the cuticle are secreted by hypodermal and seam cells and are assembled in distinct layers (Cox et al. 1981b; Edgar et al. 1982; Page and Johnstone 2007). The cuticle secreted by the hypodermal syncytium forms circumferential ridges called annuli separated from one another by furrows (Page and Johnstone 2007; Sundaram and Pujol 2024). In adults, an inner basal layer contains 2 fibrous sublayers angled in opposite directions and an outer cortical layer (Edgar et al. 1982). A fluid-filled medial layer contains hollow, nanoscale struts

built from the BLI-1, BLI-2, and BLI-6 collagens (Edgar et al. 1982; Adams et al. 2023). All layers are composed of extensively cross-linked collagens. The cortical layer also contains cuticlins, proteins that remain in the insoluble fraction after cuticle solubilization (Ristoratore et al. 1994). The cortical layer is covered by the epicuticle, a poorly understood structure that is thought to be a lipid bilayer covered by a glycoprotein rich surface coat (Blaxter 1993; Peixoto and De Souza 1995; Juarez et al. 2019). There can be stage-specific variations in cuticle structure. Only adults have a medial layer, and both L1 and adult cuticles contain alae, lateral longitudinal ridges secreted by seam cells in the epidermis (Edgar et al. 1982; Katz et al. 2022). Dauer larvae, which are a specialized stress-resistant alternative L3 stage, have alae and a thicker cuticle than other stages and express unique collagens (Cox et al. 1980; Cox and Hirsh 1985).

During each larval stage animals must build a new aECM underneath the old one and separate the old aECM (apolysis) which is subsequently shed (ecdysis) (Lažetić and Fay 2017; Sundaram and Pujol 2024). A specialized, transient structure known as the precuticle is thought to pattern the new cuticle and is then endocytosed (Cohen and Sundaram 2020). Precuticle components include zona pellucida proteins, lipocalins, fibrillin, extracellular leucine-rich repeat proteins, and hedgehog-related proteins (Gill et al. 2016; Forman-Rubinsky et al. 2017;

Cohen et al. 2019; Flatt et al. 2019; Cohen and Sundaram 2020; Serra and Sundaram 2021). The sheath is a similar structure in embryos which ensures embryonic membrane integrity and directs force during elongation (Priess and Hirsh 1986; Costa et al. 1997; Kelley et al. 2015; Vuong-Brender et al. 2017). The vulval aECM has recently been shown to be highly dynamic, and specialized aECMs also line the rectum, excretory system, mouth, and glial socket cells (Gill et al. 2016; Cohen et al. 2019; Cohen et al. 2020; Kamal et al. 2022).

A major question is how the new aECM is constructed and the old one is shed during molting? A poorly understood genetic oscillator is thought to coordinate the expression of cuticle components and processing enzymes (Hendriks et al. 2014; Meeuse et al. 2020 ). The molting cycle must also be tightly linked to developmental progression with homologs of circadian rhythm proteins (LIN-42/PER and NHR-23/ROR) playing important roles in both molting (Kostrouchova et al. 1998, 2001; Monsalve et al. 2011; Spangler et al. 2025) and developmental timers (Jeon et al. 1999; Tennessen et al. 2006; Patel et al. 2022; Kinney et al. 2023). Studies of NHR-23-regulated genes revealed an enrichment in predicted protease and protease inhibitors (Kouns et al. 2011; Johnson et al. 2023). Proteases are required for molting or ecdysis in both *C. elegans* and parasitic nematodes, presumably by promoting apolysis, though some are thought to function in collagen processing (Thacker et al. 1995; Davis et al. 2004; Hashmi et al. 2004; Frand et al. 2005; Stepek et al. 2010b, 2011; Kim et al. 2011; Birnbaum et al. 2023). Proteins with Kunitz protease inhibitor domains have been implicated in molting through RNAi screening and have been suggested to suppress ecdysis (Frand et al. 2005; Stepek et al. 2010a; Lažetić and Fay 2017). MLT-11 is a predicted protease inhibitor in the Kunitz family, and *mlt-11* RNAi causes molting defects (Frand et al. 2005). The aECM morphology mutant *rol-9* was recently mapped to the *mlt-11* gene, and the canonical *sc148* allele causes a semidominant right roller phenotype through an in-frame deletion in a conserved exon (Rich et al. 2022). *mlt-11* mRNA oscillates, peaking midmolt (Hendriks et al. 2014; Meeuse et al. 2020), and its expression is regulated by NHR-23 (Frand et al. 2005). Yet *mlt-11* remains poorly characterized.

Here, we identified 2 NHR-23-regulated *cis*-regulatory elements important for *mlt-11* expression. Deletion of both elements or *mlt-11* RNAi caused developmental delay, motility defects, a defective cuticle barrier, and aberrant localization of the collagens BLI-1 and ROL-6. *mlt-11* RNAi consistently produced stronger phenotypes than *cis*-element deletion mutants. *mlt-11(RNAi)* animals displayed defective localization of aECM components in the basal, medial, and cortical layers. Structure–function analysis using precise deletions revealed a striking range of phenotypes. Depending on the sequence deleted, we observed embryonic lethality, right rollers, left rollers, or small detachments of the cuticle that we call microblisters (µBli). mNeonGreen::3xFLAG insertions at the C-terminus and internally between Kunitz domains 2 and 3 produced distinct localization patterns. Western blot analysis suggested that MLT-11 is posttranslationally processed into at least 2 fragments. Internal MLT-11::mNG::3xFLAG displayed dynamic localization to the aECM in larvae and embryos and was endocytosed into lysosomes, like precuticle factors. In contrast, a C-terminal MLT-11:mNG::3xFLAG exhibited weaker aECM localization and a more diffuse localization in embryos and the vulva. Embryonic lethality arose from a loss of cell junction integrity during elongation. This is the first work to define the *mlt-11* null phenotype and to show that Kunitz domains and other conserved sequences may confer functional specificity.

## Materials and methods
### Strains and culture
*C. elegans* were cultured as originally described (Brenner 1974), except that the worms were grown on Modified Youngren's, Only Bacto-peptone (MYOB) media instead of NGM. MYOB agar was made as previously described (Church et al. 1995). We obtained N2, CB6193 (Partridge et al. 2008), EG7968 (Frøkjær-Jensen et al. 2014), IG274 (Pujol et al. 2008), NM5179 (Nonet 2020), and NM5548 (Nonet 2023) from the Caenorhabditis Genetics Center. Genotypes are provided in Supplementary Table 1. The names and genotypes of strains generated for this study are also provided in Supplementary Table 1.

### Other strains
JDW655 *cut-2(wrd233[cut-2::mNG::3xFLAG])* V and PHX4625 *col-19::mNG(cyb4625) X* are described elsewhere (Ragle et al. 2025). JDW913 was created by generating a *rol-6::mNG::3xFLAG* knock-in as previously described (Johnson et al. 2023) in a JDW909 *dpy-10(syb4556 wrd383[dpy-10::mScarlet]) II* background. JDW909 will be described elsewhere. XW18042 *qxIs722(dpy-7p::DPY-7::sfGFP)* was created by the lab of Xiaochen Wang (Miao et al. 2020) and provided to us by Prof. Meera Sundaram.

### Genome editing
All plasmids used are listed in Supplementary Table 2. Annotated plasmid sequence files are provided in Supplementary File 1. Sequence files for knock-ins, promoter deletions, and coding sequence deletions are provided in Supplementary File 2. Wormbase was used in the design of genome editing reagents (Wormbase 2024). Specific cloning details and primers used are available upon request. JDW380 *jsTi1493 {mosL loxP [wrdSi72(mlt-11p(-2.8 kb)::mNeonGreen(dpi)::tbb-2 3'UTR)] FRT3::mosR} IV* was created by recombination-mediated cassette exchange (RMCE) (Nonet 2020). A 2.8-kb *mlt-11* promoter fragment was initially Gibson cloned into the *NLS::mScarlet (dpi)::tbb-2 3'UTR* vector pJW1841 (Ashley et al. 2021) to generate pJW1934. The mScarlet cassette was then replaced with mNeonGreen (dpi) to generate pJW2229. The *mlt-11p (−2.8 kb) mNeonGreen (dpi)-tbb-2 3' UTR* fragment was PCR amplified from pJW2229 and Gibson cloned into *SphI-HF + SpeI-HF* double-digested RMCE integration vector pLF3FShC to produce pJW2337. This vector was integrated into NM5179, and the self-excising cassette (SEC) was excised from the strain as previously described (Dickinson et al. 2015).

A pJW2361 *mNeonGreen(dpi)::3xFLAG::PEST-tbb-2 3'UTR* vector for SapTrap with ATG and GTA connectors was constructed by first linearizing pJW2322 (Clancy et al. 2023) by PCR. We then Gibson cloned in PCR-amplified *mNeonGreen (dpi)::3xFLAG* from pJW2172 and a *linker::PEST::tbb-2 3'UTR* from pJW1836 (Ashley et al. 2021). The pJW2286 *mlt-11p (−2.8 kb)* promoter for SapTrap cloning was previously described (Clancy et al. 2023). The *mlt-11* promoter fragment from Frand et al. (2005) and the *mlt-11p (−5.3 kb)* promoter fragment were PCR amplified from a fosmid containing the *mlt-11* gene and Gibson cloned into linearized pJW2286 to make pJW2451 and pJW2457, respectively. These *mlt-11* promoter plasmids were SapTrap cloned with pJW2361 and pNM4216 to generate insertion vectors for rapid RMCE (Schwartz and Jorgensen 2016; Nonet 2023). The remaining promoter reporters were constructed by SapTrap cloning and rapid RMCE (Schwartz and Jorgensen 2016 ; Nonet 2023). A pJW2365 *pes-10 minimal promoter::mNeonGreen(dpi)::3xFLAG::PEST-tbb-2 3' UTR* vector for SapTrap with ATG and GTA connectors was constructed by linearizing pJW2361 and Gibson cloning in a *pes-10*

minimal promoter amplified from pJW1947 (Ashley et al. 2021). We chose candidate *cis-regulatory* elements through NHR-23 ChIP-seq peaks called by the modENCODE bioinformatics analysis and additional areas of open chromatin identified through Assay for Transposase-Accessible Chromatin using sequencing (ATAC-seq) (Gerstein et al. 2010; Serizay et al. 2020). The ATAC-seq and NHR-23 ChIP-seq peak DNA sequences were PCR amplified from pJW2337 or pJW2457 and Gibson cloned into pDONR221 with TGG and ATG connectors for SapTrap. These plasmids were then SapTrap cloned with pJW2365 and pNM4216 to generate insertion vectors for rapid RMCE (Schwartz and Jorgensen 2016; Nonet 2023).

For the *mlt-11* internal knock-in (JDW541), the mNeonGreen::3xFLAG cassette was inserted in an unstructured region of exon 7. This strain was created by injection of RNPs (700-ng/µL Integrated DNA Technologies (IDT) Cas9, 115-ng/µL crRNA, and 250-ng/µL IDT tracrRNA) and a dsDNA repair template (25 to 50 ng/µL) created by PCR amplification of a pJW2172 plasmid template into N2 animals (Paix et al. 2014, 2015) (Supplementary Table 2). PCR products were melted to boost editing efficiency, as previously described (Ghanta and Mello 2020). crRNAs used and repair template oligos for deletions are provided in Supplementary Table 2. F1 progeny were screened by mNeonGreen expression.

*mlt-11* promoter region deletion strains were created by injection of Cas9 ribonucleoprotein complexes (RNPs) (Paix et al. 2014; Paix et al. 2015) (700-ng/µL IDT Cas9, 115-ng/µL each crRNA, and 250-ng/µL IDT tracrRNA), oligonucleotide repair template (110 ng/µL), and pSEM229 coinjection marker (25 ng/µL) (El Mouridi et al. 2020) for screening into N2 or JDW541. Where possible, we selected "GGNGG" crRNA targets as these have been the most robust in our experience and support efficient editing (Farboud and Meyer 2015). F1s expressing the coinjection marker were isolated to lay eggs and screened by PCR for the deletion. Genotyping primers are provided in Supplementary Table 2. All deletions were confirmed by Sanger sequencing.

*mlt-11* coding sequence deletion strains were created by injection of Cas9 RNPs (Paix et al. 2014; Paix et al. 2015) [(700-ng/µL IDT Cas9, 115-ng/µL each crRNA, and 250-ng/µL IDT tracrRNA) or (250-ng/µL IDT Cas9, 20-ng/µL each crRNA, and 40-ng/µL IDT tracrRNA)], oligonucleotide repair template (110 ng/µL) and pSEM229 coinjection marker (25 ng/µL) (El Mouridi et al. 2020) for screening into strain EG7968. crRNA target selection, F1 progeny screening, isolation, and genotyping were similar to our promoter region deletion workflow. We balanced the homozygous lethal mutations genetically by crossing to a strain with a *myo-2p::GFP::unc-54 3'UTR* cassette inserted into F46B3.7, a gene roughly 40 kb away. Homozygous viable lines were not crossed to the balancing strain. Genotyping primers are provided in Supplementary Table 2.

## Imaging

Synchronized animals were collected by either picking or washing off plates. For washing, 1000 µL of M9 + 2% gelatin was added to the plate or well, agitated to suspend animals in M9 + gelatin, and then transferred to a 1.5-mL tube. Animals were spun at $700 \times g$ for 1 min. The media was then aspirated off, and animals were resuspended in 500-µL M9 + 2% gelatin with 5-mM levamisole. A total of 12 µL of animals in M9 + gel with levamisole solution was placed on slides with a 2% agarose pad and secured with a coverslip. For picking, animals were transferred to a 10-µL drop of M9 + 5 mM levamisole on a 2% agarose pad on a slide and secured with a coverslip. Images were acquired using a

Plan-Apochromat 40×/1.3 Oil DIC lens or a Plan-Apochromat 63×/1.4 Oil DIC lens on an AxioImager M2 microscope (Carl Zeiss Microscopy, LLC) equipped with a Colibri 7 light-emitting diode (LED) light source and an Axiocam 506 mono camera. Acquired images were processed through Fiji software (version: 2.0.0-rc-69/1.52p). For direct comparisons within a figure, we set the exposure conditions to avoid pixel saturation of the brightest sample and kept equivalent exposure for imaging of the other samples.

## RNAi knockdown

RNA interference experiments were performed as in Johnson et al. (2023). Control RNAi used an empty L4440. The *mlt-11* RNAi vector was streaked from the Ahringer library (Kamath et al. 2003).

## Hoechst staining

Hoechst 33258 staining was performed as described previously (Moribe et al. 2004), except that we used 10 µg/mL of Hoechst 33258 as previously described (Ward et al. 2014). Two biological replicates were performed examining 50 animals per experiment. Representative images were taken with equivalent exposures using a 63× Oil DIC lens, as described in the imaging section.

## Results

### *mlt-11* is expressed in embryonic and larval epidermal cells throughout development

To explore *mlt-11* expression, we created a single-copy *mlt-11p::mNeonGreen* (*mlt-11p::mNG*) promoter reporter using 2.8 kb of sequence upstream of the transcription start site (TSS). The reporter was expressed in embryos starting at the bean stage in posterior epithelial cells. Expression persisted through the 3-fold stage, spreading more anteriorly (Fig. 1a). We also detected expression in epidermal cells in both larvae and adults (hyp7 syncytium and seam cells), similar to previous reports using an extrachromosomal array-based promoter reporter (Fig. 1b) (Frand et al. 2005). We also detected expression in rectal and vulval cells (Fig. 1b). A previous promoter reporter used 3 kb of sequence upstream but separated from the TSS by about 2 kb of sequence (Frand et al. 2005). We created a single-copy *mlt-11p::mNeonGreen::3xFLAG::PEST* promoter reporter containing this 3 kb of sequence as well as an additional reporter containing sequence spanning from the TSS to 5.3 kb upstream (Fig. 1c). Similar expression timing and intensity were seen in these 2 reporters compared to our original 2.8-kb promoter reporter (Fig. 1c), suggesting that both shorter promoter reporters captured key *cis-regulatory* elements.

### NHR-23 regulates *mlt-11* transcription by associating with multiple regions of the *mlt-11* promoter

We next set out to identify *cis-regulatory* elements in the overlapping sequence of our 2.8-kb promoter reporter and the 5.3- to 2-kb reporter. There are 4 NHR-23 ChIP-seq peaks in the *mlt-11* promoter (Gerstein et al. 2010; Johnson et al. 2023), and the sequences under these peaks are highly conserved in other nematodes (Fig. 2a; see Conservation track). There are also single NHR-23 peaks in the gene body and 3' UTR, which we did not pursue further. There is a strong NHR-23 ChIP-seq peak (peak 3) contained in both the 5.3- to 2-kb promoter reporter and our 2.8-kb promoter reporter (Fig. 2a). The 5.3- to 2-kb reporter also contained NHR-23 ChIP-seq peak 4 (peak 4), and our reporter also contained NHR-23 ChIP-seq peaks 1 and 2 (Fig. 2a). We therefore tested whether the

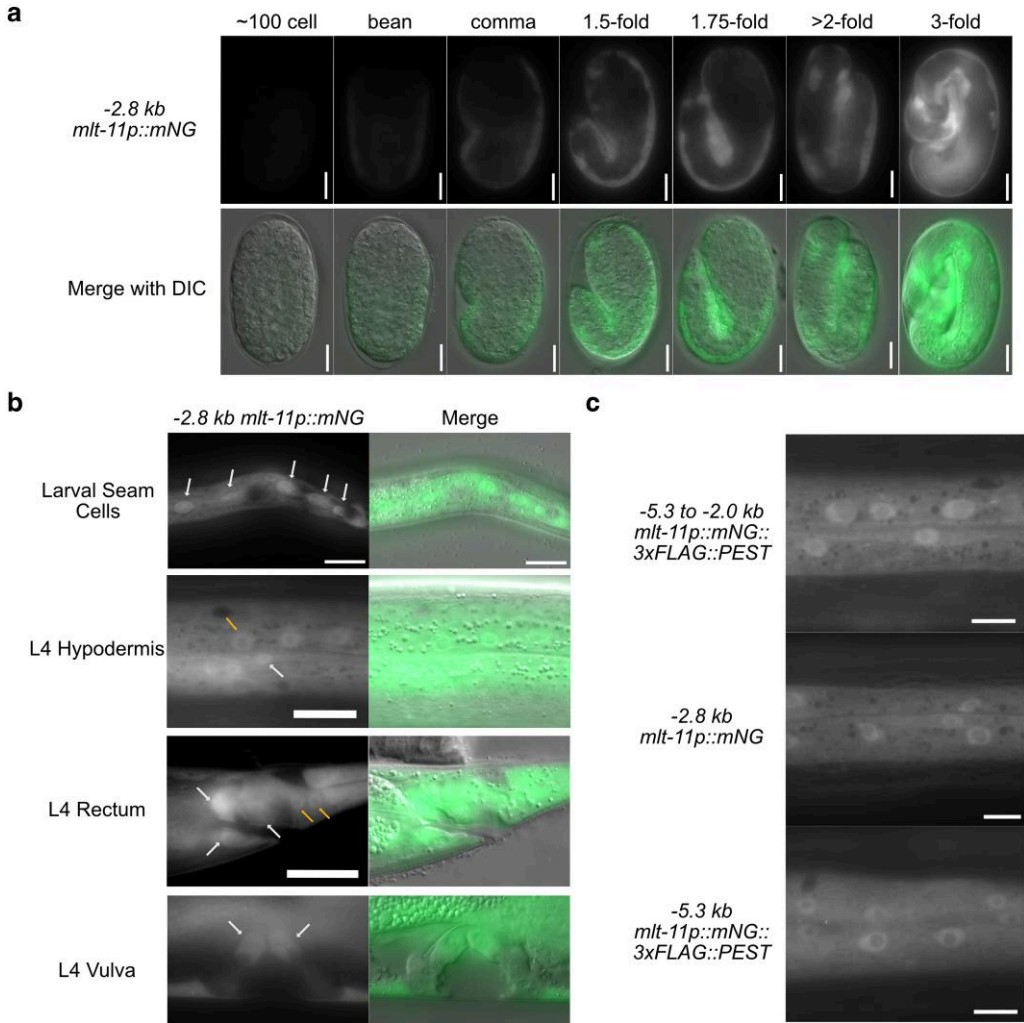

**Fig. 1.** *mlt-11* is expressed in larval and embryonic epidermal cells. a) Time course of −2.8 kb *mlt-11p::mNG::tbb-2 3′UTR* expression in embryos with stages determined by embryo morphology. A minimum of 20 embryos were observed for each developmental point over 2 experiments. b) Expression pattern of a −2.8 kb *mlt-11p::mNG::tbb-2 3′UTR* promoter reporter in hypodermal cells of L4 worms. White arrows indicate seam cells (larval seam cells and L4 hypodermis), rectal epithelial cells (L4 rectum), and vulval cells (L4 vulva). Yellow arrows indicate hypodermal cells in the external body cuticle (L4 hypodermis) and near the rectum (L4 rectum). Images are representative of 40 animals examined over 2 biological replicates. c) Hypodermal expression pattern in L4s of 3 single-copy integrated promoter reporters. The −5.3 to −2.0 kb reporter uses the same sequence as that in Frand et al. (2005), and the other 2 use 2.8 and 5.2 kb of sequence upstream of the *mlt-11* TSS. Images are representative of 40 animals examined over 2 biological replicates. Scale bars are 10 µm for embryos in (a) and 20 µm in (b and c).

sequence contained in each NHR-23 ChIP-seq peak, as well as 2 candidate enhancers identified by ATAC-seq, was sufficient to drive the expression of a reporter containing a *pes-10* minimal promoter. The *pes-10Δ::mNeonGreen* transgene displayed undetectable expression in the absence of an added *cis*-regulatory element (Fig. 2b and c). When the sequences from peaks 3 and 4, respectively, were added, we detected reporter expression in hypodermal cells with the peak 3 reporter having the most robust expression (Fig. 2b and c). Additionally, reporters that contain the sequences from peaks 3 and 4 were expressed in seam cells (Fig. 2b and c). We could not detect expression of reporters containing the sequences from peak 1 nor from 2 ATAC-seq peaks (Fig. 2b and c). The peak 2 reporter displayed very low expression in seam cells though this was close to background levels (Fig. 2b and c). To test whether NHR-23 regulated expression of the *mlt-11 peak 3 pes-10::mNeonGreen* promoter reporter, we performed *nhr-23* RNAi, which reduced expression of the full-length *mlt-11p (−5.2 kb)::mNeonGreen::3xFLAG::PEST* promoter reporter (Fig. 2d

and e) as well as *pes-10Δ::mNeonGreen* reporters containing *mlt-11* peak 3 and peak 4 (Fig. 2d and e). These data indicate that NHR-23 regulates *mlt-11* and that the DNA sequences in peaks 3 and 4 play an important regulatory role.

## DNA sequences in peaks 3 and 4 are necessary for endogenous MLT-11 expression

We next wanted to test whether the NHR-23-occupied regions from the ChIP-seq dataset were necessary for endogenous MLT-11 expression and to generate mutants with a range of MLT-11 expression. To monitor MLT-11 levels, we knocked an *mNeonGreen::3xFLAG* cassette into the 3′ end of the seventh *mlt-11* exon, producing an internal translational fusion that labels all described *mlt-11* isoforms (Fig. 3a). The strain did not display any *mlt-11* inactivation phenotypes, indicating that the knock-in did not disrupt MLT-11 function. MLT-11::mNeonGreen::3xFLAG (MLT-11::mNG(int)) was observed in the aECM of hyp7 and seam cells at the intermolt. Localization above seam cells began at L4.3 and disappeared by L4.6. In hyp7,

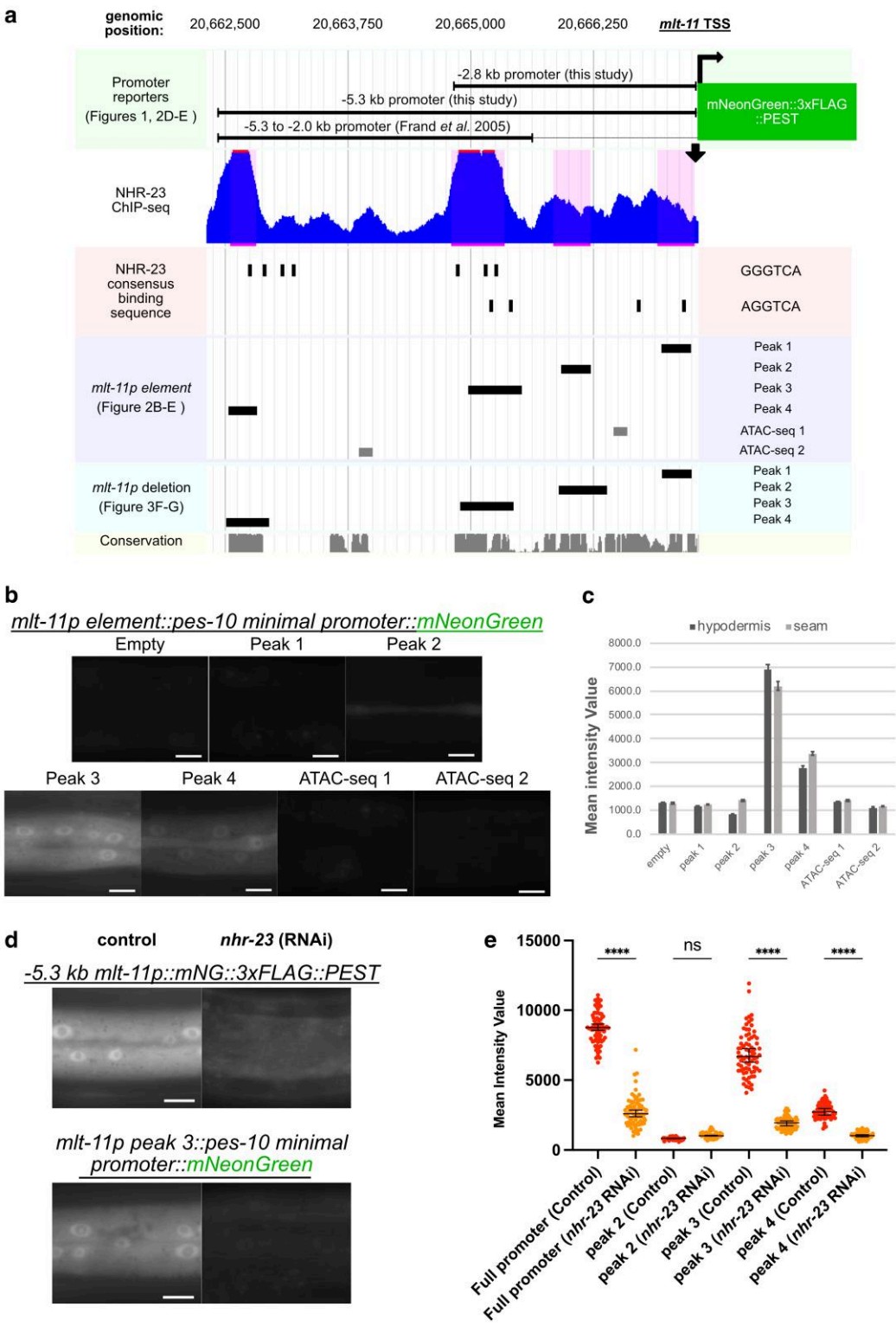

**Fig. 2.** NHR-23-occupied regions in the *mlt-11* promoter are sufficient to drive reporter expression. a) Genome browser track of relative genomic locations of *mlt-11* promoter reporters used in this study. The genomic location of NHR-23 ChIP-seq peaks, NHR-23 binding motifs, and candidate *cis*-regulatory elements in the *mlt-11* promoter are indicated. The relative genomic locations of deletions made and tested in the *mlt-11* promoter in this study and conservation calculated across 26 nematode species are also shown. b) Hypodermal expression pattern in L4s of single-copy integrated promoter reporters with the *pes-10* minimal promoter elements combined with *mlt-11* promoter sequence fragments corresponding to either NHR-23 ChIP-seq peaks or ATAC-seq open chromatin regions. c) Quantification of intensity measured in hypodermal or seam cells for each promoter reporter in (b). A total of 100 µm$^2$ boxes were drawn over hypodermal cells using the Zen image processing program and the intensity of pixels in the boxes measured. Statistical significance determined by 1-way ANOVA using GraphPad Prism 10. A minimum of 20 worms were measured for each condition over 2 independent experiments. d) Hypodermal expression pattern in L4 larvae of the indicated single-copy integrated promoter reporters. Reporter strains were fed either control or *nhr-23* RNAi bacteria. e) Quantification of intensity measured in hypodermal cells for each promoter reporter included and not shown in (d). A total of 100 µm$^2$ boxes were drawn over hypodermal cells using the Zen image processing program and intensity of pixels in the boxes measured. Statistical significance determined by 1-way ANOVA using GraphPad Prism 10. A minimum of 20 worms were measured for each condition over 2 independent experiments. Scale bars are 20 µm in (b and d).

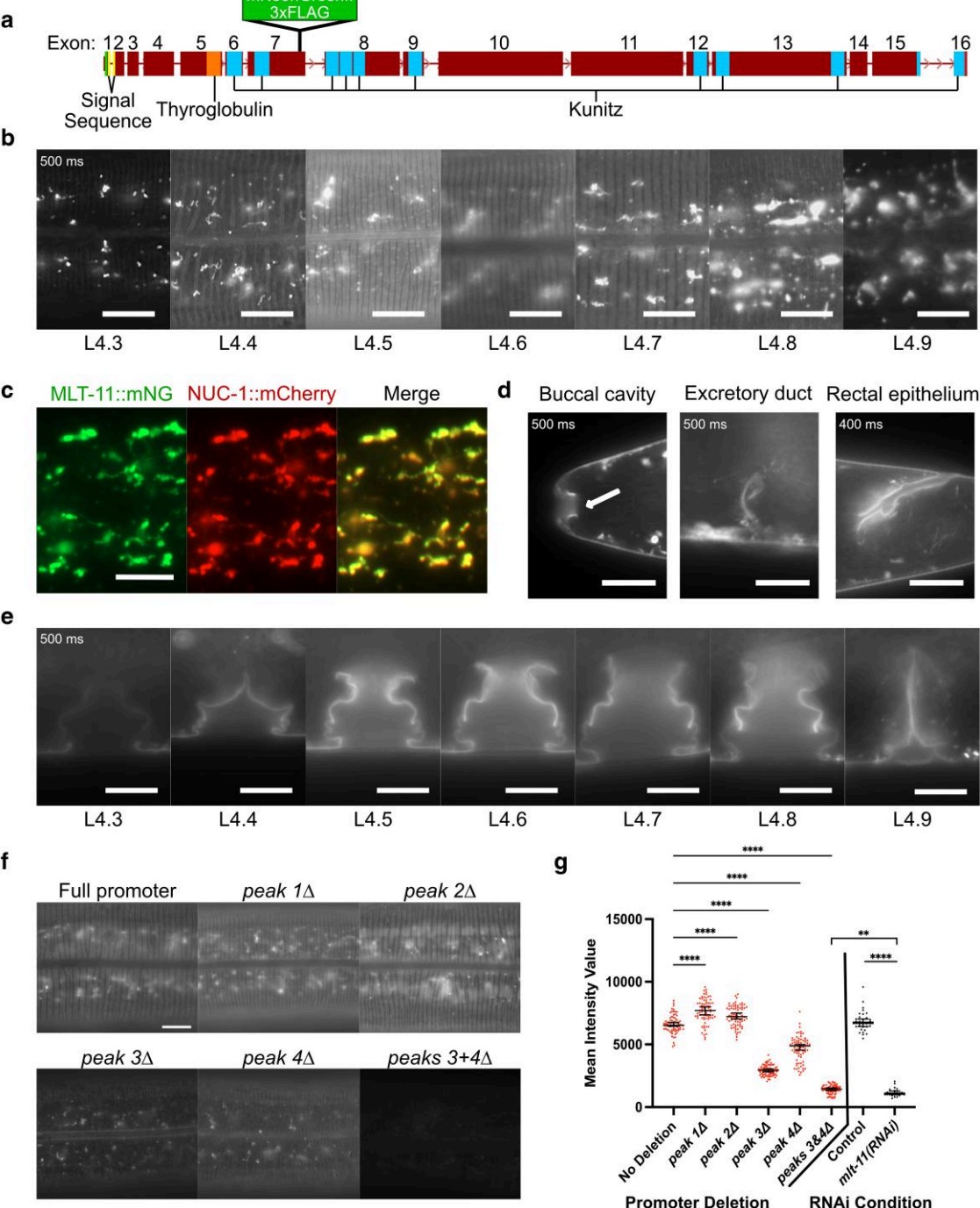

**Fig. 3.** MLT-11 localizes to the aECM in seam and hypodermal cells and interfacial orifices. a) Schematic of *mlt-11* genetic locus introns and exons, important protein domains, and the internal *mlt-11::mNG::3xFLAG* knock-in. b) Representative images of MLT-11::mNG(int) localization in hypodermal and seam cells through L4 substages. c) Colocalization of MLT-11::mNG(int) with a NUC-1::mCherry fusion protein in lysosomes of late L4 stage larvae. d) Representative images of MLT-11::mNG localization in the lumen of the buccal cavity, excretory duct, and rectum. Images are representative of a minimum of 20 animals. e) Representative images of vulval aECM localization of MLT-11::mNG through L4 substages. Images are representative of a minimum of 20 animals over 2 independent experiments. f) Representative images of MLT-11::mNG in mid-stage L4 larvae in control animals (full promoter) and with endogenous deletion of the indicated promoter regions. g) Quantification of expression of fusion proteins in this figure in the indicated genetic background or following the indicated RNAi feeding. A total of 100 µm² boxes were drawn over hypodermal cells using the Zen image processing program and intensity of pixels in the boxes measured. Statistical significance determined by 1-way ANOVA using GraphPad Prism 10. A minimum of 75 (promoter deletion) or 30 (RNAi treated) measurements were made for each condition over 2 independent experiments. Scale bars are 20 µm in (b and c) and 10 µm in (d to f).

MLT-11::mNG(int) localized first at L4.3 in thin lines reminiscent of furrows, which then transitioned to 2 parallel lines in L4.4 separated by a gap, and finally in annuli in L4.5, disappearing prior to the subsequent molt (Fig. 3b). MLT-11::mNG(int) was detected consistently

throughout larval development in hypodermal lysosomes, as determined by vesicle morphology/size (Miao et al. 2020) and colocalization with a lysosomal NUC-1::mCherry fusion protein (Fig. 3c; Clancy et al. 2023). MLT-11::mNG(int) was also observed in the

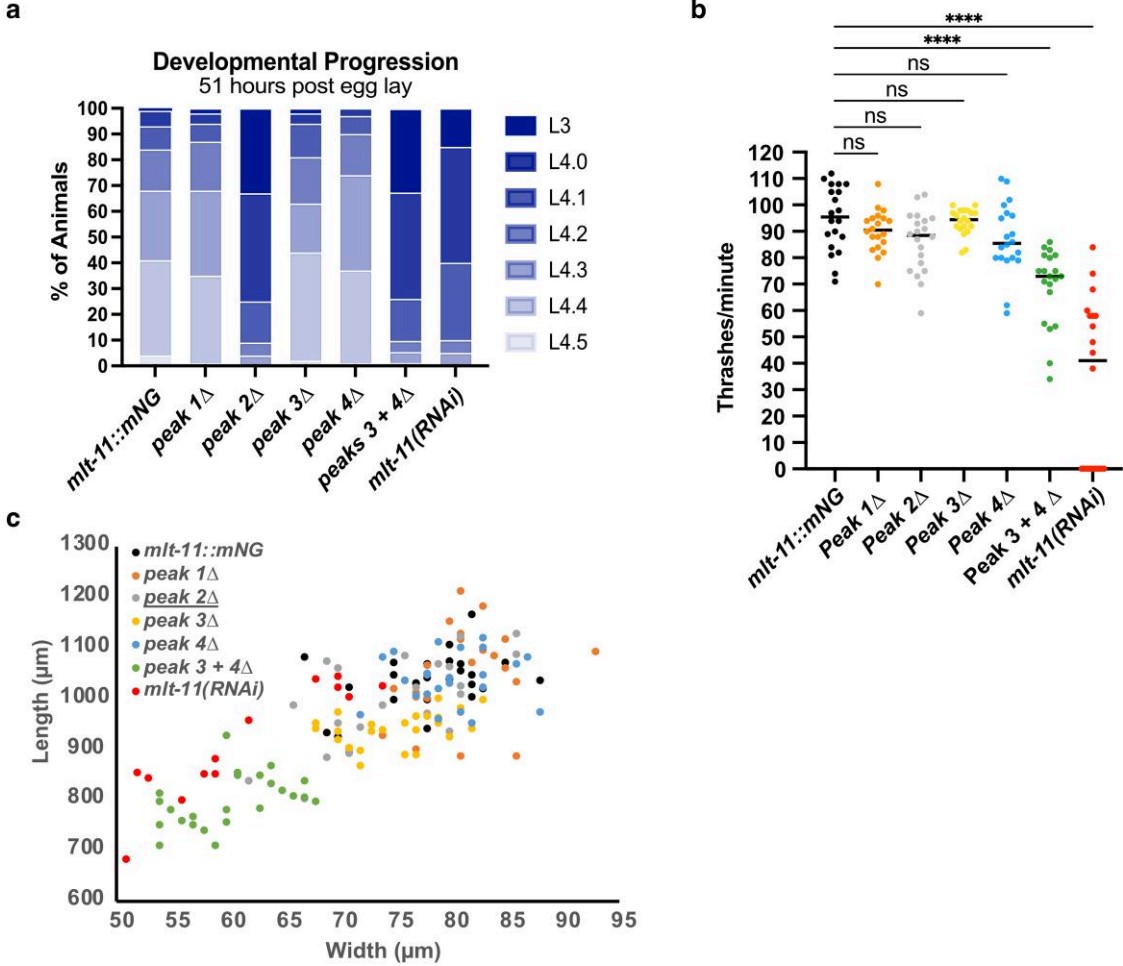

**Fig. 4.** *mlt-11* expression abrogation causes developmental delay, motility defects, and smaller bodies. Worms of the indicated genotypes and treatment (control or *mlt-11* RNAi) were allowed to lay eggs for 2 h and then removed from the plates. Embryos were allowed to develop for 51 h and then scored for a) developmental stage (L3 or L4 substage), b) thrashes per minute, and c) body length and width. Measurements represent a minimum of 50 (a) or 20 (b and c) worms over 2 experiments. Statistical significance determined by 1-way ANOVA using GraphPad Prism 10.

epithelial aECM of other external-facing orifices: the buccal cavity, vulva, excretory duct, and rectum (Fig. 3d and e).

Using CRISPR/Cas9, we individually deleted the sequences from each NHR-23 ChIP-seq peak in the MLT-11::mNG(int) background and observed expression of the translational fusion at substage L4.5 (Fig. 3f and g). We observed that the expression of MLT-11::mNG(int) was comparable to the control following deletion of the sequences from peak 1 or 2 but was reduced with deletion of the sequences from peaks 3 and 4. Simultaneous deletion of the sequences from peaks 3 and 4 in the same strain resulted in severely reduced MLT-11::mNG(int) expression, suggesting that these sequences are both necessary for full activation of expression of *mlt-11* by NHR-23. Comparing the expression reduction in the *peak 3 + 4Δ* mutant to *mlt-11(RNAi)* animals indicated that RNAi produced a significantly greater reduction in MLT-11::mNG(int) levels (Fig. 3g). We also scored developmental rate, motility, and size (Fig. 4). Interestingly, *peak 2Δ*, *peak 3 + 4Δ*, and *mlt-11(RNAi)* animals exhibited comparable developmental delay (Fig. 4a), but *peak 2Δ* animals had wild-type motility and size (Fig. 4b and c). In contrast, age-matched *peak 3 + 4Δ* and *mlt-11(RNAi)* animals exhibited a significant motility defect, with the RNAi causing a stronger phenotype, and a comparable smaller body size (Fig. 4b and c).

## Reduction of *mlt-11* expression causes defective cuticle function and structure

The developmental delay and molting defects caused by *mlt-11 (RNAi)* were reminiscent of our study of *nhr-23* (Johnson et al. 2023). As NHR-23 depletion causes a defect in the permeability barrier, we tested whether *mlt-11* inactivation also compromises this barrier. To test the cuticle barrier function, we incubated control, promoter deletion mutants, and *mlt-11(RNAi)* animals with the cuticle-impermeable, cell-membrane permeable Hoechst 33258 dye and scored animals with stained nuclei. In wild-type control and single-peak deletion animals, we observed no Hoechst staining, while we observed staining in *peak 3 + 4Δ* deletion mutants and *mlt-11(RNAi)* worms (Fig. 5a). This barrier defect was comparable to the *bus-8*-positive control strain (Partridge et al. 2008). Additionally, both *peak 3 + 4Δ* and *mlt-11(RNAi)* animals expressed an *nlp-29p::GFP* promoter reporter generally activated by infection, physical damage, or furrow collagen inactivation (Fig. 5b) (Pujol et al. 2008; Dodd et al. 2018; Martineau et al. 2021). Together, these data indicate that *mlt-11* is necessary for the barrier function of the cuticle.

Given the barrier defect observed in *mlt-11(RNAi)* and *peak 3 + 4Δ* animals and that knockdown of *mlt-11* by RNAi (Frand et al. 2005) causes molting defects, we reasoned that *mlt-11* could have a role

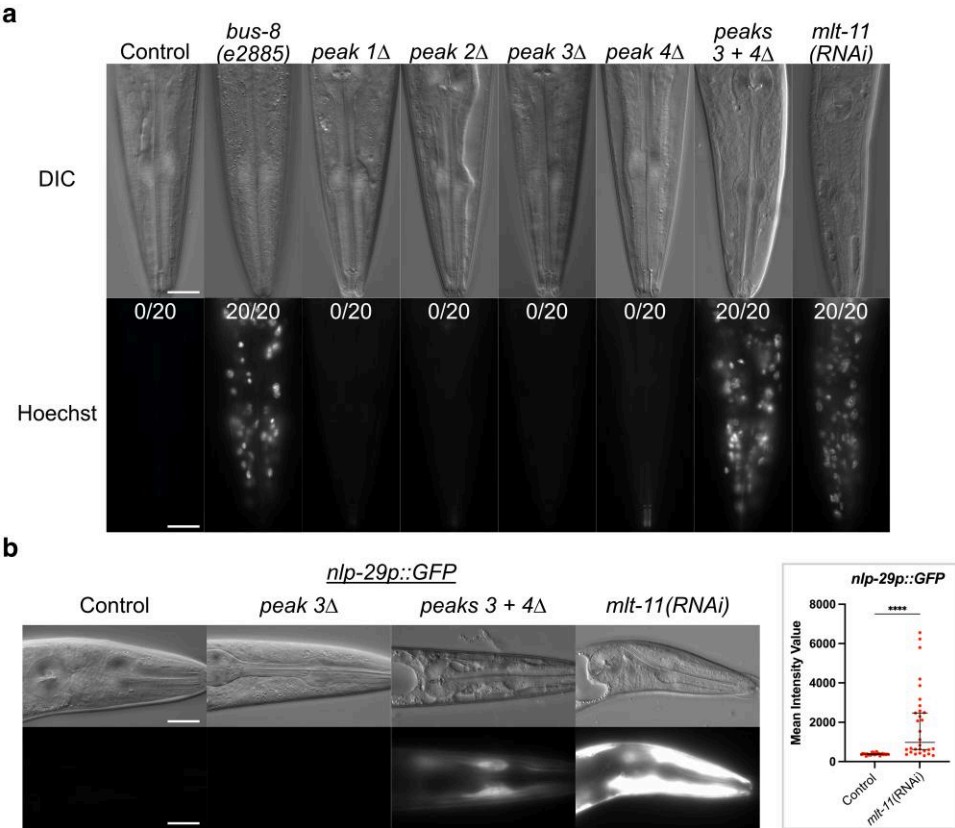

**Fig. 5.** *mlt-11* knockdown causes defective aECM structure and function. Representative images of mid-stage L4 larvae in animals of the indicated genotype/treatment (a and b). a) Worms were washed and incubated with the cuticle-impermeable, membrane-permeable Hoechst 33258 dye before being imaged. A minimum of 20 worms were scored. b) Worms carried an *nlp-29p::GFP* reporter activated by infection, acute stress, and physical damage to the cuticle (Pujol et al. 2008; Zugasti and Ewbank 2009). Three independent experiments were performed and over 40 animals scored. A total of 100 μm² boxes were drawn over hypodermal cells using the Zen image processing program, and the intensity of pixels in the boxes was measured. Statistical significance determined by 1-way ANOVA using GraphPad Prism 10. Scale bars are 20 μm.

in establishing or maintaining the structure of the cuticle. We depleted *mlt-11* by RNAi or deleted promoter elements in strains carrying translational mNG fusions to proteins that mark furrows (DPY-7 and DPY-10), the basal layer (COL-19 and ROL-6), the medial layer (BLI-1), and the cortical layer (CUT-2). We examined localization in late L4 animals (Fig. 6a) (Peixoto and De Souza 1995; Peixoto et al. 1998; McMahon et al. 2003; Adams et al. 2023; Ragle et al. 2025).

COL-19::mNG is a basal layer marker and adult-specific collagen that normally localizes to alae and annuli, but following *mlt-11* RNAi, we observed a reduction in expression and a loss of localization over the seam cells in L4 + 1 d worms (Fig. 6b and h) (Thein et al. 2003). ROL-6 is a basal layer collagen that localizes throughout L4 immediately adjacent to and flanking radial bands (furrows) (Fig. 6c) (Kim et al. 2010; Johnson et al. 2023). DPY-10 and DPY-7 are members of a class of furrow collagens (DPY-2, DPY-3, DPY-7, DPY-8, DPY-9, and DPY-10) (McMahon et al. 2003; Dodd et al. 2018; Sandhu et al. 2021) that are necessary for maintaining the furrow structure of the cuticle, proper adhesion of the cuticle to the epidermis (Aggad et al 2023), and permeability barrier function of the cuticle. ROL-6::mNG also resides in tight patches along the junction of opposing annuli above seam cells (Fig. 6c). Following deletion of *mlt-11* peak 3 or peaks 3 + 4, ROL-6::mNG localized to both furrows and annuli through L4 (Supplementary Fig. 1a). Small but distinct gaps were observed in annuli as ROL-6::mNG localized there. Seam patches were still seen, but smaller and fewer in number. Following RNAi knockdown of *mlt-11*, annuli were punctuated with larger and more

numerous gaps (Fig. 6d). Localization of ROL-6::mNG was absent over seam cells, and opposing annuli at the junction were completely separated from each other and often discontinuous at their termini (Fig. 6d). DPY-10::mScarlet exhibited branched and multidirectional furrow localization as well as short and discontinuous stretches of cuticle over seam cells (Fig. 6d). These stretches and the longitudinal bands were surrounded by thick clumps of ROL-6::mNG. DPY-7::mNG localized to furrows like DPY-10::mNG in a wild-type background and exhibited a similar branching pattern in mid-to-late L4s following *mlt-11* knockdown, suggesting that *mlt-11* inactivation broadly disrupts furrow collagen localization (Fig. 6d and e). There was a slight but significant increase in DPY-10::mScarlet and DPY-7::GFP expression following *mlt-11* RNAi (Fig. 6h). BLI-1 is a structural collagen found in the medial layer of the adult cuticle, providing connections between the basal and cortical layers in the form of vertical struts (Tong et al. 2009; Adams et al. 2023). BLI-1::mNG localized in punctae organized in circumferential rows across the hyp7 cuticle but is absent over seam cells in late stage L4s (Fig. 6f). Deletion of *mlt-11* peak 3 led to gaps between the rows and variability in the size of punctae (Supplementary Fig. 1b). Deletion of peaks 3 and 4 together or RNAi knockdown caused further disorganization of BLI-1::mNG rows and mislocalization of punctae in the cuticle above seam cells (Supplementary Fig. 1b; Fig. 6f). Similarly, CUT-2, a cuticlin in the cortical layer (Lassandro et al. 1994; Ristoratore et al. 1994), had localization patterns that were altered following knockdown of *mlt-11*. CUT-2::mNG displayed aberrant

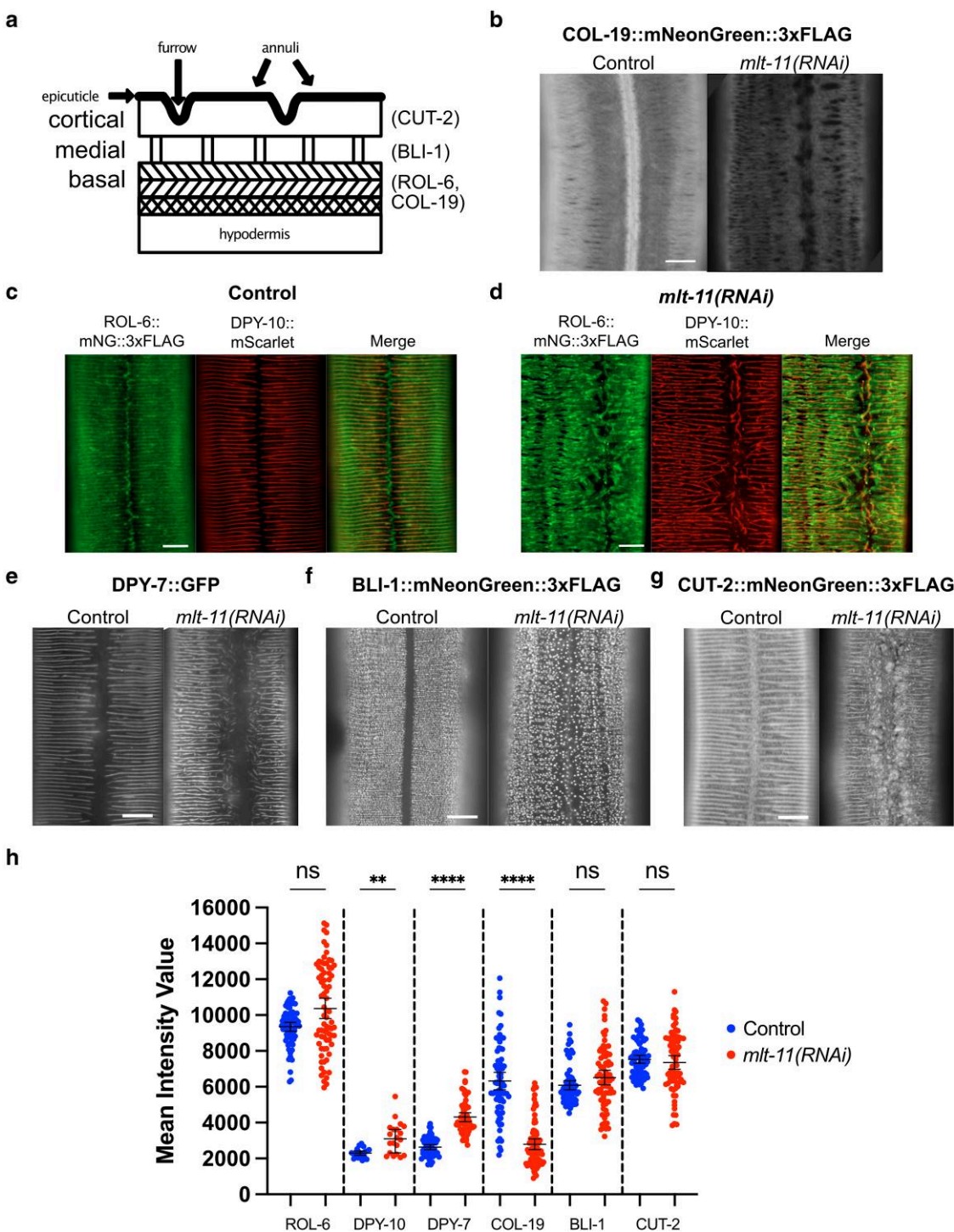

**Fig. 6.** Disruption of *mlt-11* expression by RNAi or promoter deletion causes abnormalities in the cuticle structure. a) Cartoon model of the *C. elegans* adult cuticle sublayers (epicuticle, cortical, medial, and basal) and topological features (annuli and furrows). The localization of some proteins used to mark specific layers of the cuticle in this study is indicated (CUT-2, BLI-1, ROL-6, and COL-19) (Ristoratore et al. 1994; Peixoto et al. 1998; Adams et al. 2023). b) Representative images of mNG::3xFLAG fused to COL-19 in the cuticles of young adults in control or *mlt-11*(RNAi) backgrounds. c and d) Representative images of ROL-6::mNG::3xFLAG and DPY-10::mScarlet in control or *mlt-11*(RNAi) backgrounds. e to g) Representative images of DPY-7::GFP, BLI-1::mNG::3xFLAG, and CUT-2::mNG::3xFLAG in the cuticles of mid-stage L4 larvae in control or *mlt-11*(RNAi) backgrounds. h) Quantification of expression of fusion proteins in this figure. A total of 100 μm² boxes were drawn over hypodermal cells using the Zen image processing program, and the intensity of pixels in the boxes was measured. A minimum of 20 worms were measured for each condition over 2 independent experiments, and the images are representative of the observed phenotype and variation in severity due to RNAi penetrance. Scale bars are 10 μm in (b to g).

localization over annuli and a fibrous pattern over seam cells in late L4s following *mlt-11* RNAi (Fig. 6g). Together, these data suggest that *mlt-11* is necessary for proper formation or patterning of multiple layers of the adult cuticle.

Given the aberrant localization of COL-19, DPY-10, ROL-6, DPY-7, BLI-1, and CUT-2 over the seam cells, we next examined alae morphology. The lateral alae are cuticle ridges formed by the interaction between the actin cytoskeleton in epithelial cells

and an extracellular provisional matrix (Cox et al. 1981a; Katz et al. 2022). Three continuous longitudinal ridges span the midline of lateral surfaces of adult worms (Supplementary Fig. 1c). *mlt-11(peak 3 + 4Δ)* animals had discontinuous alae, and *mlt-11* RNAi produced a more severe phenotype. Together, these data indicate that *mlt-11* is necessary for accurate development of cuticle structures derived from both hypodermal and seam cells.

## Conserved sequences in MLT-11 play distinct roles in cuticle development

MLT-11 is predicted to be a large protein (234 to 341 kDa) with a signal sequence, a thyroglobulin domain, 3 Lustrin domains, and 10 Kunitz protease inhibitor domains (Fig. 7a). A key feature of Kunitz domains are 6 conserved cysteine residues that form 3 disulfide bonds critical for stabilizing the domain (Ranasinghe and McManus 2013), and 9 of the MLT-11 Kunitz domains contain all 6 cysteines. Kunitz domain 8 is missing cysteines in the second and fourth position like Conkunitzin-S1, a functional neurotoxin in the venom of the cone snail *Conus striatus* (Supplementary Fig. 2) (Bayrhuber et al. 2005). To gain insight into MLT-11 structure and function, we generated a deletion series to determine which conserved sequences (Supplementary Fig. 3) were necessary for MLT-11 function. The thyroglobulin domain, Kunitz domains 9 to 10, a predicted furin cleavage site, and all 3 Lustrin domains appeared dispensable for development as deletion animals were viable with no overt phenotypes (Fig. 7a). Homozygous deletion of the signal sequence, the entire *mlt-11* genetic locus, Kunitz domains 2 to 10, 3 to 10, or 7 to 10 caused embryonic lethality (Fig. 7a; Supplementary Fig. 4). There was no evidence of haploinsufficiency as we could maintain balanced deletion strains. Additionally, these balanced worms produced roughly 25% arrested embryos, a rate expected for a homozygous lethal mutation (Supplementary Fig. 4). These data implicate the signal sequence and Kunitz domains 7 to 10 as being essential for embryonic development.

Animals with deletions spanning Kunitz domains 1 to 2, 3 to 5, and 3 to 6 were completely viable, producing no dead eggs as homozygotes, but instead had readily apparent (K1 to 2Δ) or mild (K3 to 5Δ and K3 to 6Δ) right roller phenotypes (Fig. 7a and b). These phenotypes are consistent with the in-frame deletion in *rol-9(sc148)* that removes one of the conserved Kunitz 2 cysteines as well as downstream sequence (Rich et al. 2022). The K1 to 2Δ roller phenotype was stronger than that of *rol-9(sc148)*. Like *rol-9(sc148)*, the K1 to 2Δ roller phenotype was detectable in heterozygotes, indicating that they are both dominant alleles, though the severity of the roller phenotype was more pronounced in homozygotes. In the milder roller alleles (K3 to 5Δ and K3 to 6Δ), we could not detect rolling heterozygotes. Surprisingly, deletion of the large genetic region between Kunitz domains 6 and 7 caused a fully penetrant left roller phenotype (Fig. 7a and b).

The deletion of Kunitz 8 or the region spanning Kunitz domains 7 to 8 caused very small and regular separations of the cortical layer from the basal layer we have termed μBli (Fig. 7c, green arrows). We observed these subtle separations across the entire cuticle from head to tail. These μBlis are larger and more frequent in worms with combined deletion of the regions spanning Kunitz 7 to 8 and Kunitz 9 to 10 (Fig. 7c, yellow arrow), suggesting that these domains contribute to this facet of cuticle integrity. Connections between the cortical and basal layers were still maintained in these animals (Fig. 7c, white arrows) but lost in many cases where much larger blisters formed (Fig. 7c, blue arrow). To ensure these cuticle deformities were separations of distinct layers within the cuticle and not separations of the entire cuticle from the

hypodermis as observed in furrow collagen mutants (Aggad et al. 2023), we expressed a ROL-6::mNG fusion in the Kunitz 7 to 8/Kunitz 9 to 10 double deletion background (Fig. 7d). The fusion protein localized to the basal layer immediately adjacent to the hypodermis but was absent from the cortical layer, suggesting that this connection was intact and these cuticle disruptions were blisters. To elucidate the fate of medial layer connections in these μBli worms, we expressed a BLI-1::mNG fusion in the Kunitz 7 to 8/Kunitz 9 to 10 double deletion strain (Fig. 7e). This led to an enhancement of the phenotype with blisters becoming larger and more penetrant suggesting that the *bli-1::mNG* could be a sensitized background. BLI-1::mNG foci were seen embedded in both the cortical (Fig. 7e, white arrow) and basal (Fig. 7e, yellow arrow) sublayers of the separated cuticle. The unique phenotypes resulting from deletion of distinct Kunitz domains suggest specific roles for these domains during cuticle development.

## MLT-11 is processed, and the C-terminal fragment displays distinct localization and dynamics

Given the distinct phenotypes produced by domain deletions in the N- and C-terminal portion of the protein, we returned to a C-terminal translational fusion that we had previously generated (MLT-11::mNG(C-term)) (Fig. 8a) (Clancy et al. 2023). Our previous analysis of this strain suggested that MLT-11::mNG(C-term) was proteolytically cleaved and predominantly localized to lysosomes with weak expression in the rectum, vulva, and cuticle (Clancy et al. 2023). We also detected localization to other interfacial matrices, such as the cuticle lining the opening of the pharynx and the excretory duct (Fig. 8b). Interestingly, we observed diffuse localization in the vulval lumen (Fig. 8b), which sharply contrasts with the localization of the internal MLT-11::mNG to apical surfaces of vulval cells (Fig. 3e). To gain insight into MLT-11 processing, we performed a western blot time course (Fig. 8c). We detected weak MLT-11::mNG(C-term) expression in early L4 and then strong expression of both a full-length product and a processed fragment by mid-L4 (Fig. 8c). Given the ~31.5 kDa size of the mNG::3xFLAG tag, the band size is consistent with a 20- to 30-kDa MLT-11 C-terminal fragment produced by cleavage between Kunitz domains 8 and 9. By late L4, the bulk of product detected was cleaved MLT-11 and levels then became undetectable as MLT-11::mNG(C-term) is not expressed in adults.

## *mlt-11* is essential for embryogenesis

Internal and C-terminal MLT-11::mNG translational fusions display distinct vulval localization patterns, Kunitz domains 7 to 10 are essential for embryogenesis, and MLT-11 is processed into at least 2 fragments. We therefore performed a time course to examine internal and C-terminal MLT-11::mNG localization over embryonic development. Internal MLT-11::mNG localization was observed in the embryonic sheath beginning in the bean stage and covered the entire animal until shortly before hatching. At this point, localization in punctae internal to the hypodermis and reminiscent of lysosomes was observed (Fig. 9a, yellow arrows). C-terminal MLT-11::mNG expression localized to the embryonic sheath in the bean stage and in the extraembryonic space between the animal and the egg case (Fig. 9b, blue arrows). This localization persisted until just prior to hatching, at which point MLT-11::mNG(C-term) was detected in the gut of many embryos (Fig. 9b, white arrow) and in internal hypodermal punctae (Fig. 9b, yellow arrow). These data suggest that N- and C-terminal regions of MLT-11 have differing expression and localization patterns.

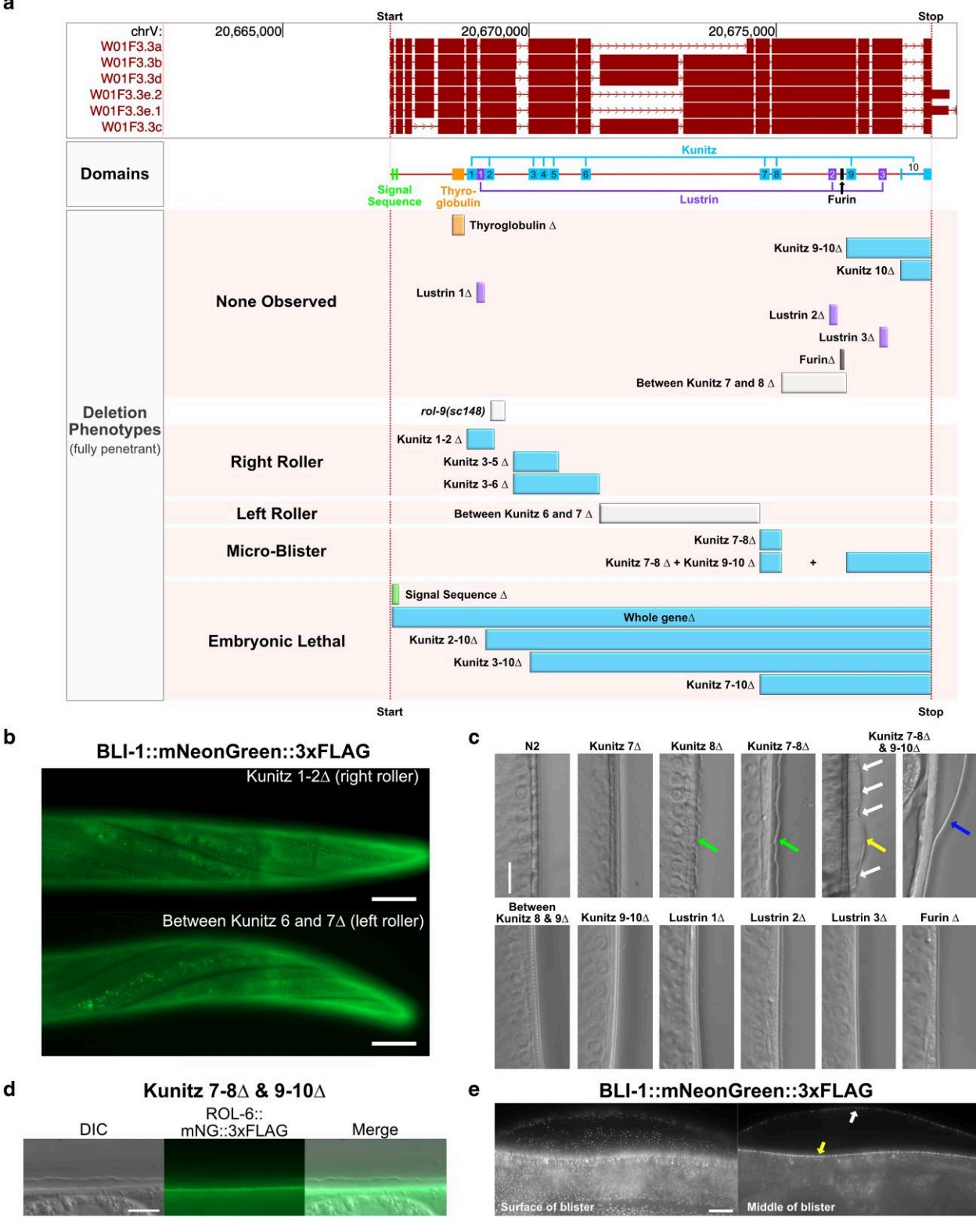

**Fig. 7.** Kunitz domains in MLT-11 with sequence similarity contribute uniquely to cuticle development. a) Schematic of the UCSC Genome Browser track showing the isoforms, domains, deletions, and associated phenotypes in the *mlt-11* genomic locus. b) Representative images of BLI-1::mNG::3xFLAG expression and localization in a Kunitz 1 to 2Δ right roller and in a strain harboring a deletion between Kunitz 6 and 7 left rollers. c) Representative images of various deletion strains showing fully intact cuticles (most strains), small μBli (green arrows), medium μBli (yellow arrows), cortical layer connections maintained with the basal layer through medial struts (white arrows), and larger blister (blue arrows). d) Representative images of DIC and basal layer ROL-6::mNG::3xFLAG in Kunitz 7 to 8Δ + Kunitz 9 to 10Δ double deletion background. e) Representative images of BLI-1::mNG::3xFLAG foci in the surface of a larger blister, the basal layer adjacent to the hypodermis, and the cortical layer that has separated from the basal layer. b to e) Images represent a minimum of 40 worms from a minimum of 3 independent experiments. Scale bars are 20 μm in (b) and 10 μm in (c to e).

**Fig. 8.** MLT-11 is processed and the C-terminal fragment displays distinct localization and dynamics. a) Schematic of *mlt-11* genetic locus introns and exons, important protein domains, and the internal and C-terminal *mlt-11::mNG::3xFLAG* knock-in. b) Representative images of MLT-11::mNG localization in the buccal cavity, excretory duct, hypodermal/seam cells, vulva lumen, and rectal epithelium. Images are representative of 40 animals examined from a minimum of 3 independent experiments. Scale bars are 10 µm in all images except for the hypodermal panel, which is 20 µm. c) Immunoblotting with the indicated antibodies of *mlt-11::mNG::3xFLAG* lysates harvested at the indicated time points postrelease. A developmental stage for each time point as determined by vulva morphology is provided (Mok et al. 2015). The blot is representative of 2 independent experiments.

The localization dynamics of MLT-11(int) were reminiscent of precuticle components such as NOAH-1, which are required for maintaining embryo integrity during elongation (Vuong-Brender et al. 2017). We therefore investigated the integrity of cell membranes using a DLG-1::mNG allele to mark adherens junctions (Heppert et al. 2018). In control embryos, DLG-1::mNG labeled adherens junctions in the pharynx, intestine, and hypodermis (Fig. 9c). In contrast, stage-matched C-terminal Kunitz deletion embryos displayed severe disorganization similar to that seen upon inactivation of the precuticle components *noah-1* and *noah-2* (Vuong-Brender et al. 2017) (Fig. 9c). The pharynx and foregut adherens junctions appeared wild type, but the hypodermal junctions were disorganized, and there was evidence of invaginations and severe disorganization in the hypodermis (Fig. 9c). *mlt-11* RNAi in an *nlp-29p::GFP* promoter reporter also caused embryonic lethality, suggesting that the reporter strain might be a sensitized background. *mlt-11*(RNAi) embryos also activated the *nlp-29p::GFP* reporter, suggesting damage to the developing cuticle (Fig. 9d). Together these data suggest that MLT-11 is necessary for the late stages of embryogenesis.

## Discussion

Despite their importance, how aECMs are built and dynamically remodeled during development and disease remains poorly understood. Using the *C. elegans* cuticle as a model aECM, we show that *mlt-11* is directly regulated by the NHR-23 transcription factor. MLT-11::mNG(int) transiently localizes to the embryonic and larval aECM before endocytosis and trafficking to a compartment that is most likely lysosomal. MLT-11::mNG(int) also lines openings to the exterior (vulva, buccal cavity, excretory duct, and rectal epithelium). *mlt-11* inactivation affects all 3 layers of the adult cuticle, indicating a broad role for MLT-11 in patterning this aECM. In agreement with these broad defects in aECM component localization, *mlt-11* inactivation causes loss of the epithelial barrier and activation of an epidermal damage reporter. MLT-11 is processed into at least 2 fragments, and internal and C-terminal mNG knock-ins display distinct localization patterns. Alleles predicted to cause a loss of MLT-11 function cause embryonic lethality with severe disorganization of hypodermal adherens junctions. Together, this work suggests that MLT-11 acts similarly to precuticle

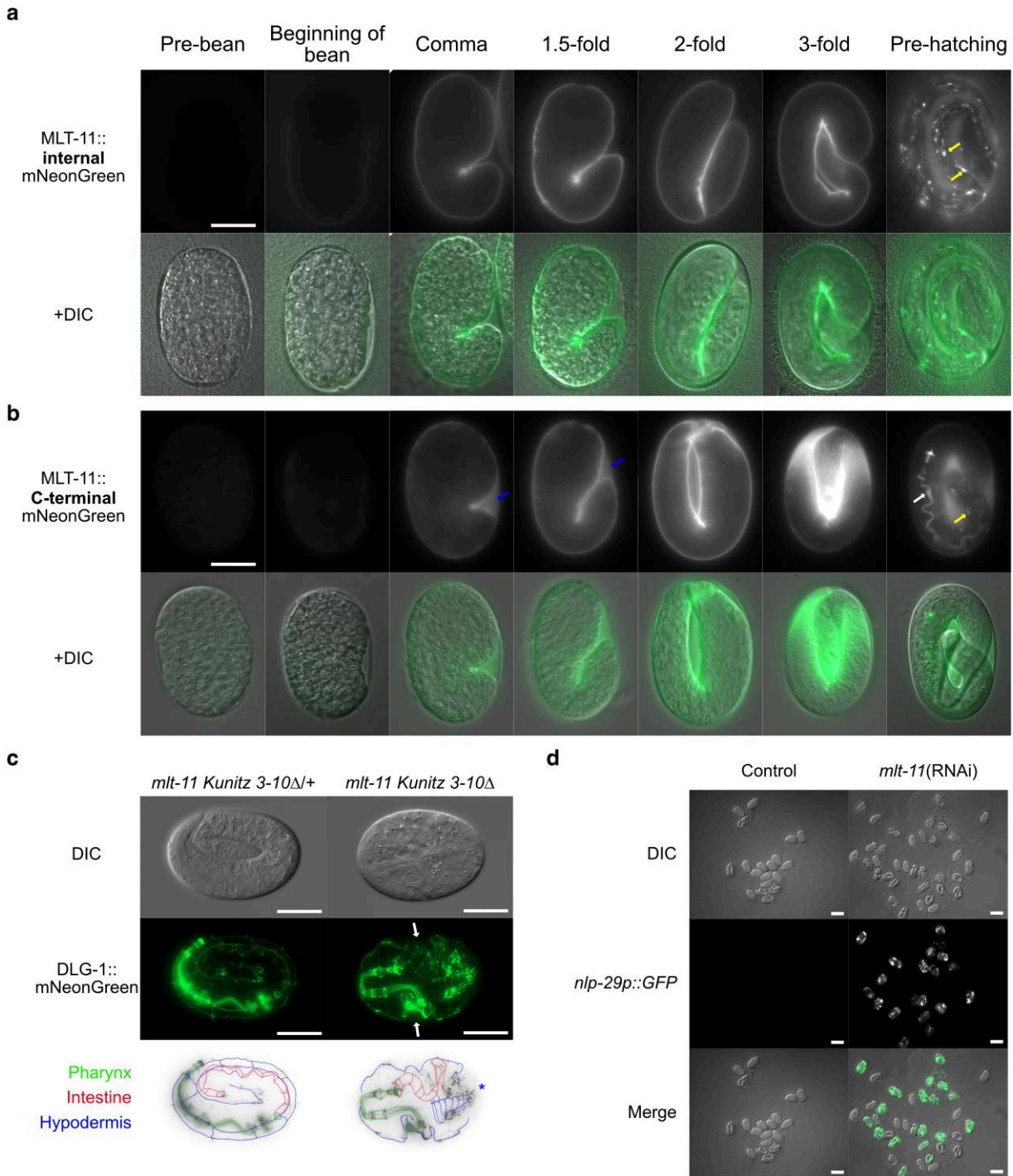

**Fig. 9.** *mlt-11* is an essential gene required for embryogenesis. Time course of internal or C-terminal MLT-11::mNG expression, respectively, in embryos with stages determined by embryo morphology (a and b). Yellow arrows indicate localization to globular structures in hypodermal cells reminiscent of lysosomes. White arrow indicates C-terminal MLT-11::mNG in the lumen of the intestine. Images represent over 20 embryos for each developmental time point in multiple observations. Blue arrows indicate C-terminal MLT-11::mNG in the extraembryonic space between the organism and the egg case. c) DIC and DLG-1::mNG images of embryos laid by a *mlt-11(Kunitz 3 to 10Δ)* heterozygous animal. One-quarter of the embryos show the disrupted DLG-1 pattern on the right. Images are representative of 40 animals examined over 2 biological replicates. White arrows show invaginations of the outer membrane in homozygous mutant embryos. Blue star indicates the suspected aggregation of epithelial cells in homozygous mutant embryos. Scale bars are 20 μm. d) Images of *nlp-29p::GFP* embryos from young adult worms plated on control or *mlt-11* RNAi bacteria. Images represent 2 biological replicates. Scale bars are 50 μm.

components, promoting patterning of all layers of the cuticle and maintaining epithelial integrity during embryonic elongation.

## Structure–function analysis suggests Kunitz domains confer functional specificity

As *mlt-11* null alleles cause embryonic lethality (Fig. 7a; Supplementary Fig. 4), we explored whether deletion of putative *cis*-regulatory elements in the *mlt-11* promoter could provide viable reduction-of-function alleles. *mlt-11* is an NHR-23-regulated gene with 4 potential NHR-23 regulatory elements in the promoter (Frand et al. 2005; Kouns et al. 2011; Johnson et al. 2023) (Fig. 2). Consistent with the reporter data (Fig. 2), deletion of the endogenous peak sequences revealed that peaks 3 and 4 produced the strongest effect on MLT-11::mNG(int) expression, and a double

deletion of peaks 3 and 4 resulted in even stronger reduction in expression (Fig. 3). Comparing *cis*-regulatory element deletion mutants to *mlt-11(RNAi)* animals revealed a range of phenotypic severity. ROL-6::mNG and BLI-1::mNG localization appeared to be the most sensitive to MLT-11 levels, as we saw aberrant localization for both in the *peak 3Δ* mutant. The only other phenotype that we observed for a single-peak deletion mutant was the curious developmental delay of the *peak 2Δ* mutant, which displayed no other defects. Interestingly, a promoter reporter for this *cis*-element was expressed in the seam cells (Fig. 2), and tissue-specific RNAi experiments indicate that NHR-23 activity is particularly important in seam cells (Johnson et al. 2023). An important future direction is to explore the tissue specificity of MLT-11 activity and the roles of the seam cells and hypodermis in molting.

Peak 3 + 4 deletion caused a significant reduction in MLT-11 levels (Fig. 3) and resulted in comparable developmental delay, smaller body size, and cuticle barrier defects like those caused by *mlt-11* RNAi (Figs. 4 and 5), suggesting that these phenotypes are next most sensitive to MLT-11 levels. Cuticle barrier defects, *nlp-29p::GFP* reporter activation, and alae gaps were caused by both peak 3 + 4 deletion and *mlt-11* RNAi, though stronger phenotypes were seen in *mlt-11(RNAi)* animals (Fig. 5; Supplementary Fig. 1). Ecdysis defects, specifically entrapment in partially or fully separated cuticles, were only observed following *mlt-11* RNAi, and embryonic lethality was only observed in *mlt-11* null alleles or *mlt-11* RNAi in an *nlp-29p::GFP* background, which is likely a sensitized background (Fig. 7). This indicates that very low levels of MLT-11 are required to prevent these defects.

## MLT-11 is proteolytically processed and distinct activities reside in the N- and C-termini

Western blotting revealed that MLT-11 is rapidly processed into a 20- to 30-kDa fragment, which is consistent with cleavage between Kunitz domains 8 and 9, and C-terminal and internal translational fusions displayed distinct localization patterns and dynamics (Figs. 3, 8, and 9). We have not been successful with western blots to assess internal MLT-11::mNG abundance and size, so it is unclear whether the protein is processed into additional fragments. The internal MLT-11::mNG localization was reminiscent of precuticle components, such as NOAH-1 (Vuong-Brender et al. 2017). MLT-11::mNG(int) was secreted and endocytosed in both embryos and larvae and localized to the embryonic sheath (Figs. 3 and 9). Yet curiously, deletions of Kunitz domains in this region did not produce the lethality characteristic of precuticle components (Kelley et al. 2015; Vuong-Brender et al. 2017; Sundaram and Pujol 2024). Rather, deletions of Kunitz 1 to 2, 3 to 5, and 3 to 6 produced roller phenotypes. In contrast, C-terminal deletions (Kunitz 7 to 10) produced embryonic lethality and loss of epithelial cell integrity reminiscent of *noah-1* mutants (Vuong-Brender et al. 2017). The C-terminal translational fusion displayed a similar localization to precuticle components in embryos (Vuong-Brender et al. 2017) and had a diffuse localization in both embryos and the vulva (Figs. 8b and 9a). The foregut localization of MLT-11::mNG(C-term) at the end of embryogenesis is reminiscent of BLI-4::sfGFP localization (Birnbaum et al. 2023). BLI-4 is a subtilisin/kexin family protease required for embryonic elongation, larval development, and formation of BLI-1/BLI-2 nanoscale collagen struts (Thacker et al. 1995; Adams et al. 2023; Birnbaum et al. 2023). Exploring whether MLT-11 and BLI-4 interact and whether MLT-11 impacts the BLI-4-dependent processing of collagens such as SQT-3 and DPY-17 is a logical extension of this work.

When we deleted Kunitz domains 7 to 8, we observed a novel "μBli" phenotype, in which the cortical and basal layers

periodically separated. Like the Bli phenotype of *bli-1* or *bli-2* mutants (Adams et al. 2023), the μBli was only observed in adults, suggesting that it likely arises from destabilization of BLI-1/BLI-2 struts. Notably, deleting Kunitz 7 to 8 in a *bli-1::mNG* background exacerbated the μBli phenotype, suggesting the *bli-1::mNG* was a sensitized background. Deletion of Kunitz domains 9 to 10 also exacerbated this phenotype, leading to larger microblisters. Curiously, deletion of Kunitz domains 7 to 10 causes embryonic lethality, but deletion of the intervening sequence between Kunitz domain 8 and 9 produced a superficially wild-type phenotype. It is not currently clear whether the μBli and embryonic lethality represent common molecular defects or distinct defects.

Loss of Kunitz domains 1 to 2, 3 to 5, or 3 to 6 produced right rollers. These data are consistent with the molecular identification of the semidominant right roller *rol-9(sc148)* allele as an inframe deletion in *mlt-11* that removes the last cysteine of Kunitz domain 2 as well as 67 amino acids of downstream conserved sequence (Rich et al. 2022). Kunitz 1 to 2 deletion produced a stronger semidominant roller phenotype, suggesting that the *sc148* domain may disrupt Kunitz 2 function. Testing the individual contributions of Kunitz 1 and 2 and engineering mutations predicted to disrupt Kunitz activity is a key future experiment. Surprisingly, deletion of the sequence between Kunitz domains 6 and 7, which contained conserved sequence blocks (Supplementary Fig. 3) but no predicted domains, caused a completely penetrant left roller phenotype. *mlt-11* thus appears to be in an exclusive class of "ambidextrous" roller genes in which specific mutations can produce either left or right rollers, depending on the mutation. *sqt-1* is another member of this class. Mutations in *sqt-1* that alter a conserved carboxyl domain cysteine prevent crosslinking into dimers, tetramers, and oligomers, resulting in left rollers (Yang and Kramer 1999). Mutation of a predicted subtilisin protease cleavage site causes extra sequence to be retained on the SQT-1 N-terminus and a right roller phenotype (Yang and Kramer 1999). The *mlt-11* phenotypes suggest that different Kunitz domains or conserved sequences may control interactions with specific substrates. Given the critical role of proteases such as BLI-4 and DPY-31 in collagen processing, testing for genetic interactions between *mlt-11* mutations and *bli-4* and *dpy-31* alleles is an important future direction. Additionally, testing how left and right *mlt-11* roller alleles affect the localization and processing of collagens with Rol phenotypes is another priority.

## MLT-11 is required for patterning and function of multiple cuticle layers

Mutation or depletion of oscillating collagens has been shown to affect the structural organization of collagens with similar temporal expression dynamics, suggesting that these collagens might be part of a common substructure (McMahon et al. 2003). Temporally, during the L4 stage, *mlt-11* mRNA peaks in expression close to when *bli-1* mRNA is expressed and before the *rol-6* and *cut-2* mRNA expression peak (Meeuse et al. 2020). An internally tagged MLT-11::mNG translational fusion displays aECM localization from L4.3 to L4.8 with a peak localization at L4.5 (Figure 3). Surprisingly, *mlt-11* inactivation affected the localization of proteins in the basal, medial, and cortical layers of the cuticle (Fig. 6). In the basal layer, MLT-11 was required for the wild-type localization COL-19 and ROL-6 (Fig. 6b to d). For COL-19, *mlt-11* RNAi caused a significant reduction in expression as well as a loss of localization to alae (Fig. 6b). *mlt-11* inactivation caused the formation of short fragments of the furrow collagens DPY-7 and DPY-10 that were often at aberrant angles deviating from the wild-type circumferential pattern (Fig. 6c to e). It is unclear

whether the shorter fragments reflect breakage of longer fibers or formation of shorter fibers. Curiously, DPY-10 fibers normally terminate over the seam cells, but following *mlt-11* RNAi short fibers formed in parallel to the longitudinal axis (Fig. 6c and d). ROL-6, which normally flanks furrow collagens, displayed a disorganized aggregated pattern but still appeared to flank DPY-10 fibers (Fig. 6c and d). In the medial layer, *mlt-11* inactivation resulted in larger, disorganized BLI-1 punctae that were no longer excluded from the alae region (Fig. 6f). The cortical marker CUT 2 normally forms circumferential furrow fibers and localizes over seam cells (Fig. 6g). Following *mlt-11* RNAi, we still observed furrow fibers, but they were frequently disorganized, and there was severe disorganization flanking and above the seam cells (Fig. 6g). Interestingly, NHR-23 depletion also causes aberrant ROL-6 and NOAH-1 localization over the seam cell (Johnson et al. 2023). Exploring the relative roles of seam and hypodermal cells in aECM assembly and whether these defects reflect a role for MLT-11 in patterning each layer or whether a defect in an early aECM structure caused by *mlt-11* inactivation is propagated to subsequent layers are important future questions.

*mlt-11* inactivation also produced a defective cuticle barrier and activation of an epidermal stress reporter (Fig. 5). It is not clear what is leading to the cuticle barrier defect following reduction of MLT-11 levels. RNAi screens of collagens found that only knockdown of furrow collagens caused a barrier defect (Sandhu et al. 2021) and epidermal stress reporter activation (Dodd et al. 2018), and *mlt-11* inactivation caused defective localization of the furrow collagens DPY-7 and DPY-10 (Fig. 6c to e). *bus-8* is a glycosyltransferase required for localization of the molting regulator MLT-8 to lysosomes (Wu et al. 2022). Given the localization of MLT-11 in lysosomes, it is possible that MLT-11 functions in this compartment to promote barrier integrity. Alterations in the cortical layer and epicuticle can also lead to barrier defects (Njume et al. 2022; Pooranachithra et al. 2024). The dynamic aECM localization, broad effects on cuticle structure, and role in embryonic morphogenesis are highly reminiscent of precuticle components such as NOAH-1 (Vuong-Brender et al. 2017; Cohen et al. 2020). Exploring the interrelationship between MLT-11 and precuticle components will provide insight into how the aECM is assembled.

MLT-11 has 10 predicted protease inhibitor domains, most of which contain residues necessary for Kunitz inhibitor activity (Supplementary Fig. 3) (Ascenzi et al. 2003; Ranasinghe and McManus 2013), which could buffer reduction of MLT-11 levels. The position and the number of Kunitz domains are strongly conserved out to parasitic nematodes with 300 to 500 million years of divergence from *C. elegans*. It is possible that *mlt-11* mutant and knockdown phenotypes result from aberrant protease activity. MLT-11 localizes to both the aECM and lysosomes, so phenotypes could arise from unrestrained protease activity in either compartment. Some proteases, such as BLI-4 (PCSK family), SURO-1 (zinc carboxypeptidase), and DPY-31 (astacin metalloprotease), are thought to be involved in collagen processing in the secretory pathway, while others such as NAS-36 and NAS-37 (astacin metalloproteases) promote apolysis (Thacker et al. 1995; Davis et al. 2004; Suzuki et al. 2004; Frand et al. 2005; Stepek et al. 2010b, 2011; Kim et al. 2011; Birnbaum et al. 2023; Sung et al. 2025). Interestingly, SURO-1 was discovered in a genetic screen that identified suppressors of a dominant left roller phenotype caused by overexpression of *rol-6(su1006)* (Kim et al. 2011). ADM-2, the sole *C. elegans* member of the ADM-meltrin metalloprotease family, suppresses the molting defects of *nekl* reduction-of-function alleles, which impair endocytosis (Joseph et al. 2022). The

ADAMTS protein, ADT-2, is implicated in body size control and collagen organization, which is notable given the smaller sized animals and cuticle disorganization produced by *mlt-11* inactivation (Figs. 4 and 6) (Fernando et al. 2011). Cathepsins are lysosomal proteases (Britton and Murray 2004; Miedel et al. 2012), and lysosome dysfunction can cause impaired degradation of endocytosed aECM components and molting defects (Miao et al. 2020). Kunitz domains typically inhibit serine proteases, and the above-mentioned proteases are all in other protease families. However, an atypical and functionally divergent family of Kunitz domain-containing proteins in the parasitic helminth, *Fasciola hepatica* was shown to have no activity against serine proteases but rather inhibited cysteine proteases such as cathepsins (Smith et al. 2020). Thus, it is important to consider a broad range of potential MLT-11 substrates. It is also possible that MLT-11 is acting as a scaffold or accessory factor, as a related Kunitz domain-containing protein, BLI-5, was shown to enhance the activity of porcine pancreatic elastase and bovine pancreatic alpha-chymotrypsin (Stepek et al. 2010a). Screening for genetic and protein–protein interactions between *mlt-11* and proteases is a high-priority future direction to determine how MLT-11 ensures proper cuticle structure and function.

## Data availability

A full description of all oligonucleotides, plasmids, transgenes, and *C. elegans* strains created and used in this article is in the set of supplemental tables. The authors affirm that all data necessary for confirming the conclusions of the article are present within the article, figures, tables, and Supplementary material. Plasmid sequences are provided in Supplementary File 1. Knock-in sequences from genome editing are provided in Supplementary File 2. Any additional information is provided upon request.

Supplemental material available at GENETICS online.

## Acknowledgments

The authors thank David Fay, Meera Sundaram, Andrew Chisholm, and Nathalie Pujol for their helpful conversations. Andrew Chisholm and Nathalie Pujol provided invaluable feedback on the manuscript. The authors also thank Krista Myles, Patricia Bliatout, Zoe Johnson, Javier Hernandez Lopez, Zoie Reyna, Emma Cadena, Valarie Hallin, and Olivia Vedar for their research support and Andrew Chisholm, Michael Nonet, Xiaochen Wang, and Meera Sundaram for strains. Some strains were provided by the Caenorhabditis Genetics Center, which is funded by the NIH Office of Research Infrastructure Programs (P40 OD010440). Wormbase was used in the design and execution of experiments.

## Funding

This work was funded by the National Institutes of Health (NIH) National Institute of General Medical Sciences (NIGMS) (R00GM107345, R01GM138701, and R35GM158317) to J.D.W.

## Conflicts of interest

The authors declare no competing or financial interests.

## Author contributions

James Matthew Ragle (Conceptualization, Data curation, Formal analysis, Methodology, Resources, Supervision, Validation, Writing—original draft, Writing—review & editing), Jordan

D. Ward (Conceptualization, Data curation, Formal analysis, Funding acquisition, Methodology, Project administration, Resources, Supervision, Validation, Writing—original draft, Writing—review & editing), Ariela Turzo (Methodology, Formal analysis, Validation, Writing—review & editing), Anton Jackson (Methodology, Formal analysis, Validation, Writing—review & editing), An A. Vo (Methodology, Formal analysis, Validation, Writing—review & editing), Vivian T. Pham (Methodology, Formal analysis, Validation, Writing—review & editing), Keya Daly (Methodology, Formal analysis, Validation, Writing—review & editing), John C. Clancy (Methodology, Formal analysis, Validation, Writing—review & editing), Max T. Levenson (Methodology, Formal analysis, Validation, Writing—review & editing), and Alex D. Lee (Methodology, Formal analysis, Validation, Writing—review & editing)

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

*Editor: B. Grant*