## [Peer Review File · Genetics]

MLT-11 is necessary for *C. elegans* embryogenesis and conserved sequences play distinct roles in cuticle structure

James Ragle, Ariela Turzo, Anton Jackson, An Vo, Vivian Pham, Keya Daly, John Clancy, Max Levenson, Alex Lee, and Jordan Ward

NOTE: The reviews and decision letters are unedited and appear as submitted by the reviewers.

In extremely rare instances and as determined by a Senior Editor or the EIC, portions of a review may be redacted. If a review is signed, the reviewer has agreed to no longer remain anonymous.

The review history appears in chronological order.

Review Timeline:

Submission Date:	2024-10-11
Editorial Decision:	2024-11-04
Resubmission Received:	2025-11-13
Accepted:	2025-11-30

November 4, 2024

GENETICS-2024-307539

The NHR-23-regulated putative protease inhibitor mlt-11 gene is necessary for *C. elegans* cuticle structure and function

Dear Jordan:

Two experts in the field have reviewed your manuscript, and I have read it as well. While your manuscript is not currently acceptable for publication in GENETICS, we would be happy to consider a substantially revised manuscript. Both reviewers have comments and concerns to be addressed in a revised manuscript. You can read their reviews at the end of this email.

Both reviewers recognized the overall quality of the work, while one reviewer felt that the advance would be better suited for G3. I tend to agree that the manuscript would benefit from increased scope or depth of analysis, although the quality provided by always using endogenously tagged and mutated genes and single copy transgenic reporters somewhat ameliorates this for me. The authors should seriously consider how the scope or depth could be improved, either as suggested by the reviewer, or otherwise. Both reviewers noted several figures where data quantification is needed. Biological replication and trial numbers are needed for other results. Please provide these. We look forward to receiving your revised manuscript. Please let the editorial office know approximately how long you expect to need for revisions.

Upon resubmission, please include:

1. A clean version of your manuscript;
2. A marked version of your manuscript in which you highlight significant revisions carried out in response to the major points raised by the editor/reviewers (track changes is acceptable if preferred);
3. A detailed response to the editor's/reviewers' feedback and to the concerns listed above. Please reference line numbers in this response to aid the editor and reviewers.

Your paper will likely be sent back out for review.

Additionally, please ensure that your resubmission is formatted for GENETICS
<https://academic.oup.com/genetics/pages/general-instructions>

Follow this link to submit the revised manuscript: Link Not Available

Sincerely,

Barth

Barth Grant
Associate Editor
GENETICS

Approved by:
Meera Sundaram
Senior Editor
GENETICS

Reviewer #2 :

This manuscript by Ward and colleagues examines the roles of several upstream enhancer regions within the mlt-11 locus to determine their effects on mlt-11 expression and cuticle-associated phenotypes. Prior studies by several groups indicated that mlt-11 is a cyclically regulated gene required for molting and that its expression is regulated directly by the steroid hormone receptor, NHR-23. This study examines four upstream enhancer regions, each approximately several hundred bp in size, which were previously indicated to bind NHR-23 by ChIP-seq (modencode). The current study finds evidence that these regions confer variable effects on mlt-11 expression along with cuticle organization and associated phenotypes.

The manuscript is clearly written and presented, and the chosen data is fundamentally solid with reasonable conclusions and discussion. At the same time, my personal view is that the presented story would be better suited to G3 than Genetics based on its overall scope, depth, and amount of new information. Based on my understanding, a Genetics paper might contain either an

in-depth high-resolution dive into the regulation of *mlt-11* by NHR-23 or perhaps a more comprehensive characterization of MLT-11 functions. In any case, the submitted manuscript appears to be more of a focused G3-type story.

Perhaps if the authors chose to combine the current study with some of the data described in their 2022 bioRxiv preprint on *mlt-11*, this might result in a story that would be more appropriate for Genetics. The 2022 preprint shares a similar title to the submitted manuscript and contains a good amount of overlapping (similar or identical) data, but it also contains some additional experiments not included. Specifically, I thought that the non-overlapping information from figures 2F, Fig 5, Fig 6, and Fig 7C,D would have fit quite well with the data that was presented in the current manuscript. The Kunitz domain assays and requirement for MLT-11 during embryogenesis could also be included and would broaden the scope of the study. Admittedly, how the authors partition their findings into publishable units is up to them, but this seems like one possible option.

Below are some specific comments on the submitted work. Mainly, I thought there could/should have been somewhat more quantification of certain results, which would strengthen the conclusions without (likely) requiring additional experimentation.

Major point

Some of the results in Figs 2B, 3C, and 6B-E would be strengthened with quantification. Although I believe that there are differences between the various strains and mutants, some were subtle, and most could benefit from some kind of systematic non-biased analyses. Along these lines, some of the panel backgrounds (2B and 3C) appeared to be quite different, possibly due to exposure length or post hoc corrections. At present we have only representative images for the major findings although the authors indicate that a decent number of worms were examined/imaged.

Minor points

- 1) The apparent clear-cut effects on MLT-11 expression in the *delta-1-4* deletions (3C) didn't always correlate with phenotypic outcomes. Is there a way to reconcile this?
- 2) Fig 2A. It was hard to see peaks 1 and 2 and peak 2 didn't appear to contain a consensus binding site for NHR-23. In general, it wasn't entirely clear how/why the specific regions were chosen.
- 3) "Together, this suggests *mlt-11* is necessary for proper formation or patterning of each layer of the adult cuticle." Based on the timing of *mlt-11* expression is this likely to be the case? Along those lines, "Temporally, the *mlt-11* mRNA peaks in expression close to the *bli-1* expression peak". Does this mean mid cycle? Could defects in one layer indirectly alter the others?
- 4) "One explanation for this trend is that genes that peak in expression closer to *nhr-23* could be more responsive to NHR-23 levels, similar to *E. coli* amino acid biosynthesis in which genes earlier in the pathway are more responsive" Are these two phenomena comparable? Might the operon effect on 3' genes simply be due to Pol2 processivity (falling off) versus something presumably quite different happening with NHR-23?
- 5) Any thoughts about the lysosomal expression or function of MLT-11? These puncta do look a bit like intestinal autofluorescence, although I presume that was ruled out? This group published a very useful paper to the field regarding the artifactual accumulation of RFPs in lysosomes. I assume this can't be the case given the tag is NG.

Edits

- 1) Line 50 and others. Need a space between last word in the sentence and the reference.
- 2) Line 66. "number or duration of molting" change "molting" to "molts" or "molting cycles"
- 3) Line 70. Remove comma.
- 4) Line 107. Possibly refer to 2A to provide a visual for the promoter regions mentioned.
- 5) Line 120. Suggest, "In addition, the Frand et al. reporter contained NHR-23 ChIP-seq peak 4 (Peak 4), whereas our reporter contained NHR-23..."
- 6) Line 203. Change "longitudinal to "lateral" or "circumferential"?"
- 7) Fig 3 legend. "MLT-11 also localized to the"

As for all my reviews, this is signed by David Fay

Reviewer #3 :

The authors present a careful examination of cis-promoter elements that regulate a central player in *C. elegans* molting. Molting is a defining characteristic of nematodes that is surprisingly poorly understood. Insights into this process provide understanding of oscillatory gene expression, epidermal ECM remodeling, and function of a Kunitz protease inhibitor. The authors use the latest methods for single copy insertion of large DNA inserts, testing of elements at their native positions with CRISPR Cas9, and new ECM protein reporters for different layers of the cuticle. Deleting of native elements is elegant and allowed testing for functional consequences. The writing is clear and data generally presented logically and clearly and the conclusions are supported by results. However, more details are needed for experimental design and sampling and quantification of image data is lacking.

Major comments

Quantification is lacking for results that tested the function of elements or RNAi on reporter fluorescence (e.g., Figs 2B-C, 3C, and 5A-B). Details on biological and trial replication are also needed to evaluate reproducibility (in the legends). Single copy insertions should be less variable than traditional extrachromosomal arrays, but variation and reproducibility are still important to report.

For qualitative effects (e.g., changes in patterns in Fig. 6), at the least percentages and numbers of worms tested is needed and a detailed explanation of how phenotypes were scored.

Similarly, biological replication and trial numbers are needed for other results, e.g., Fig 4B-C.

Minor comments

Lines 124-125 and 151 - How was the absence of cis elements tested? Please explain if they were deletions or scrambles.

Lines 129-130 - Please provide more explanation for what sequences were included to test the sufficiency of specific elements.

Associate Editor Comments:

We thank the reviewers for their helpful and constructive suggestions, which drastically improved this manuscript. We have provided a point-by-point response to their comments below. We have extensively revised the manuscript adding both data from previous preprints and new data. Quantitation has been added, as both reviewers requested. Furrow collagen translational reporters have been added to Figure 6 and in Figure 7 we add extensive structure-function analysis showing the distinct MLT-11 deletions can produce a range of phenotypes (left roller, right roller, microblisters, embryonic lethality). We added a C-terminal MLT-11:mNG translational fusion and demonstrated that it is rapidly post-translationally processed. The MLT-11::mNG internal and C-terminal knock-ins displayed distinct localizations in larvae and embryos. *mlt-11* null embryos display severe disorganization of epithelial adherens junctions. Note that we removed the *bli-4* genetic interaction data from the original submission. We were very excited about the connection to the BLI-4 protease, however subsequent experiments revealed that *mlt-11* is a general blister suppressor which will be a future study. We hope that these revisions address the previous concern of the reviewers and substantially increase the scope of the paper.

Reviewer #2 :

This manuscript by Ward and colleagues examines the roles of several upstream enhancer regions within the *mlt-11* locus to determine their effects on *mlt-11* expression and cuticle-associated phenotypes. Prior studies by several groups indicated that *mlt-11* is a cyclically regulated gene required for molting and that its expression is regulated directly by the steroid hormone receptor, NHR-23. This study examines four upstream enhancer regions, each approximately several hundred bp in size, which were previously indicated to bind NHR-23 by ChIP-seq (modencode). The current study finds evidence that these regions confer variable effects on *mlt-11* expression along with cuticle organization and associated phenotypes.

The manuscript is clearly written and presented, and the chosen data is fundamentally solid with reasonable conclusions and discussion. At the same time, my personal view is that the presented story would be better suited to G3 than Genetics based on its overall scope, depth, and amount of new information. Based on my understanding, a Genetics paper might contain either an in-depth high-resolution dive into the regulation of *mlt-11* by NHR-23 or perhaps a more comprehensive characterization of MLT-11 functions. In any case, the submitted manuscript appears to be more of a focused G3-type story.

Perhaps if the authors chose to combine the current study with some of the data described in their 2022 bioRxiv preprint on *mlt-11*, this might result in a story that would be more appropriate for Genetics. The 2022 preprint shares a similar title to the submitted manuscript and contains a good amount of overlapping (similar or identical) data, but it also contains some additional experiments not included. Specifically, I thought that the non-overlapping information from figures 2F, Fig 5, Fig 6, and Fig 7C,D would have fit quite well with the data that was presented in the current manuscript. The Kunitz domain assays and requirement for MLT-11 during embryogenesis could also be

included and would broaden the scope of the study. Admittedly, how the authors partition their findings into publishable units is up to them, but this seems like one possible option.

We appreciate the reviewer's thoughts on which data from our previous preprints could make this work suitable for Genetics. We have added substantial data to the original submission including both data from the previous preprints and new data.

We've added time course data to Fig 3B, demonstrating that MLT-11 transiently localizes to the aECM before being internalized, much like precuticle components. We have added DPY-7::GFP and DPY-10::mScarlet data to Fig 6 and also performed co-localization data with ROL-6::mNG and DPY-10::mScarlet. We included our *mlt-11* deletion series coupled with phenotypic analysis from our previous preprints and expanded it with additional deletion mutants.

Below are some specific comments on the submitted work. Mainly, I thought there could/should have been somewhat more quantification of certain results, which would strengthen the conclusions without (likely) requiring additional experimentation.

Major point

Some of the results in Figs 2B, 3C, and 6B-E would be strengthened with quantification. Although I believe that there are differences between the various strains and mutants, some were subtle, and most could benefit from some kind of systematic non-biased analyses. Along these lines, some of the panel backgrounds (2B and 3C) appeared to be quite different, possibly due to exposure length or post hoc corrections. At present we have only representative images for the major findings although the authors indicate that a decent number of worms were examined/imaged.

We have added quantitation and statistical comparison in Figs 2C, 2E, 3G, 5B, and 6H. We have added details on biological and trial replication where previously lacking.

Minor points

1) The apparent clear-cut effects on MLT-11 expression in the *delta-1-4* deletions (3C) didn't always correlate with phenotypic outcomes. Is there a way to reconcile this?

We have added a description of the effect on the *delta 1-4* levels on MLT-11 levels and had a broader discussion on MLT-11 levels (incorporating RNAi data and nulls) and phenotype. The outlier was the peak 2 deletion which cause really no reduction in levels but a developmental delay. We do not understand this phenotype and it's an area of future research.

2) Fig 2A. It was hard to see peaks 1 and 2 and peak 2 didn't appear to contain a

consensus binding site for NHR-23. In general, it wasn't entirely clear how/why the specific regions were chosen.

We completely appreciate this point. We have added text to the methods (Lines 633-636) describing how we chose these regulatory regions. We appreciate that peaks 1 and 2 are not very high and peak 2 lacks NHR-23 consensus binding sites., but they were called as NHR-23 peaks by the modENCODE bioinformatic analysis so we included them in our analyses. We also included two additional areas of hypodermal open chromatin identified by ATAC-seq experiments conducted by the Ahringer lab.

3) "Together, this suggests *mlt-11* is necessary for proper formation or patterning of each layer of the adult cuticle." Based on the timing of *mlt-11* expression is this likely to be the case? Along those lines, "Temporally, the *mlt-11* mRNA peaks in expression close to the *bli-1* expression peak". Does this mean mid cycle? Could defects in one layer indirectly alter the others?

We have added additional translational fusion data to Figure 6 and lines 508-534 discuss the peak of *mlt-11* expression, when the protein is detected in the aECM, and the possibility of direct vs. indirect impacts on aECM structure.

4) "One explanation for this trend is that genes that peak in expression closer to *nhr-23* could be more responsive to NHR-23 levels, similar to *E. coli* amino acid biosynthesis in which genes earlier in the pathway are more responsive" Are these two phenomena comparable? Might the operon effect on 3' genes simply be due to Pol2 processivity (falling off) versus something presumably quite different happening with NHR-23?

These are very fair points and in the major text revision this portion of the text has been cut as the manuscript now focuses more on MLT-11 function and less on regulation of expression.

5) Any thoughts about the lysosomal expression or function of MLT-11? These puncta do look a bit like intestinal autofluorescence, although I presume that was ruled out? This group published a very useful paper to the field regarding the artifactual accumulation of RFPs in lysosomes. I assume this can't be the case given the tag is NG.

Great point, we have added NUC-1 co-localization to demonstrate that MLT-11 is in lysosomes (Figure 3C) and lines 5538-546 in the discussion cover MLT-11 localization and potential sites of action.

Edits

1) Line 50 and others. Need a space between last word in the sentence and the reference.

These corrections have been made.

2) Line 66. "number or duration of molting" change "molting" to "molts" or "molting cycles"

This correction has been made.

3) Line 70. Remove comma.

This correction has been made.

4) Line 107. Possibly refer to 2A to provide a visual for the promoter regions mentioned.

This edit has been made.

5) Line 120. Suggest, "In addition, the Frand et al. reporter contained NHR-23 ChIP-seq peak 4 (Peak 4), whereas our reporter contained NHR-23..."

This edit has been made.

6) Line 203. Change "longitudinal to "lateral" or "circumferential"?"

This correction has been made.

7) Fig 3 legend. "MLT-11 also localized to the"

This correction has been made.

As for all my reviews, this is signed by David Fay

Thanks as always for the great feedback, David!

Reviewer #3 :

The authors present a careful examination of cis-promoter elements that regulate a central player in *C. elegans* molting. Molting is a defining characteristic of nematodes that is surprisingly poorly understood. Insights into this process provide understanding of oscillatory gene expression, epidermal ECM remodeling, and function of a Kunitz protease inhibitor. The authors use the latest methods for single copy insertion of large DNA inserts, testing of elements at their native positions with CRISPR Cas9, and new ECM protein reporters for different layers of the cuticle. Deleting of native elements is elegant and allowed testing for functional consequences. The writing is clear and data generally resented logically and clearly and the conclusions are supported by results. However, more details are needed for experimental design and sampling and

quantification of image data is lacking.

Major comments

Quantification is lacking for results that tested the function of elements or RNAi on reporter fluorescence (e.g., Figs 2B-C, 3C, and 5A-B). Details on biological and trial replication are also needed to evaluate reproducibility (in the legends). Single copy insertions should be less variable than traditional extrachromosomal arrays, but variation and reproducibility are still important to report.

We have added quantitation and statistical comparison in Figs 2C, 2E, 3G, 5B, and 6B-F. We have added details on biological and trial replication where previously lacking.

For qualitative effects (e.g., changes in patterns in Fig. 6), at the least percentages and numbers of worms tested is needed and a detailed explanation of how phenotypes were scored.

Similarly, biological replication and trial numbers are needed for other results, e.g., Fig 4B-C.

We have added quantitative data for changes in intensity and the number of animals scored and biological replicate number.

Minor comments

Lines 124-125 and 151 - How was the absence of cis elements tested? Please explain if they were deletions or scrambles.

We have added additional text to the methods (lines 622-630) clarifying how we deleted the sequences. The candidate cis elements were deleted by oligo-templated homologous recombination and validated by PCR and Sanger sequencing.

Lines 129-130 - Please provide more explanation for what sequences were included to test the sufficiency of specific elements.

We used the sequence of the ChIP-seq peaks from the Wormbase Genome browser and the ATAC-seq peaks from the Ahringer lab Regulatory Atlas website (<https://ahringierlab.com/RegAtlas/>). We have added this information to the methods methods (Lines 633-636).

Associate Editor Comments:

November 30, 2025

RE: GENETICS-2025-308777

Prof. Jordan D. Ward
University of California Santa Cruz
Molecular, Cell, and Developmental Biology
1156 High Street
Sinshimer 324
Santa Cruz, California 95064

Dear Jordan:

Congratulations, your manuscript titled "MLT-11 is necessary for *C. elegans* embryogenesis and conserved sequences play distinct roles in cuticle structure" is accepted for publication in GENETICS! Many thanks for submitting your research to the journal.

One reviewer had a few suggestions for improving the manuscript that you may want to consider. You can view their comments at the bottom of this email.

To Proceed to Publication:

1. Format your article according to GENETICS style: <https://academic.oup.com/genetics/pages/author-guidelines>
2. Ensure that you comply with data and community resource citation guidelines: <https://academic.oup.com/genetics/pages/author-guidelines#section-5-9-2>
3. Upload your final files at <https://genetics.msubmit.net>
4. Add oupsupport@scipris.com and genetics.oup@novatechset.com (or the domains @scipris.com and @novatechset.com) to your email program's "safe senders" list. You will be contacted by both at various points during the production process.

Notes:

- Your currently-accepted manuscript (unedited, as submitted, reviewed, and accepted) will be published at GENETICS and deposited into PubMed as an Advance Access article. Notify sourcefiles@thegsajournals.org before signing your license if you do not wish to publish your article via Advance Access.
- We invite you to submit an original color figure related to your paper for consideration as cover art. Please email your submission to the editorial office or upload it with your final files. You can submit a small-sized image for evaluation, and if selected, the final image must be a TIFF file 2513px wide by 3263px high (8.375 by 10.875 inches; resolution of 600ppi). Please avoid graphs and small type.
- After files are sent to Oxford University Press we use SciPris to manage article licensing and payment. If you do not have a SciPris account, you will receive an email from no-reply@scipris.com to sign up to use Oxford University Press' author portal. After logging in, follow the online instructions to sign your license and arrange any payment due.

If you have any questions or encounter any problems while uploading your accepted manuscript files, please email the editorial office at sourcefiles@thegsajournals.org.

Sincerely,

Barth

Barth Grant
Associate Editor
GENETICS

Approved by:
Meera Sundaram
Senior Editor

GENETICS

Review comments (if applicable):

Reviewer #2 :

The authors have done an excellent job addressing concerns and have added substantially to the manuscript, providing a detailed analysis of MLT-11 and its complex role in regulating aECM remodeling.

Two very minor things the authors may wish to double check.

- 1) Line 247. Does the author mean longitudinal or radial bands?
- 2) Line 421. Figure 7 or Figure S4?

Reviewer #3 :

All of my comments were addressed and the additional functional data broaden the relevance of the study.

Senior Editor comments:

Glad to see this accepted! I have a few suggestions for minor text edits you could make in the final version.

Consider changing the title to something less granular, e.g.: Roles and regulation of the kunitz domain protein MLT-11 during *C. elegans* cuticle synthesis and molting

lines 52-53, "Animals have a collagen-based aECM (cuticle) that may provide insight into mammalian skin biology dynamics (Page and Johnstone 2007; Sundaram and Pujol 2024)." This could be rephrased given that mammalian skin aECM is a very different keratin-based matrix.

line 246, it's confusing to call furrow collagen DPY-10 a "basal layer" collagen

Figure 1A, some embryo stage names don't match the photos - e.g. the embryo called "twitching" is 1.5-fold, the animal called 1.5-fold is 1.75-fold, the animal called 2-fold is somewhat older with tail already turned.

Figure 7 title, I think you mean they have similar sequences, but to me "genetically similar" implies they have similar functional requirements, which is the opposite of your argument.

Figure 8C, legend mentions developmental stage corresponding to these timepoints, but no stage information is shown in the figure